# Personalized Dictionary Learning for Heterogeneous Datasets

**Geyu Liang**
University of Michigan
Ann Arbor, MI 48109
lianggy@umich.edu

**Naichen Shi**
University of Michigan
Ann Arbor, MI 48109
naichens@umich.edu

**Raed Al Kontar**
University of Michigan
Ann Arbor, MI 48109
alkontar@umich.edu

**Salar Fattahi**
University of Michigan
Ann Arbor, MI 48109
fattahi@umich.edu

## Abstract

We introduce a relevant yet challenging problem named *Personalized Dictionary Learning (*PerDL*)*, where the goal is to learn sparse linear representations from heterogeneous datasets that share some commonality. In PerDL, we model each dataset's shared and unique features as *global* and *local* dictionaries. Challenges for PerDL not only are inherited from classical dictionary learning (DL), but also arise due to the unknown nature of the shared and unique features. In this paper, we rigorously formulate this problem and provide conditions under which the global and local dictionaries can be provably disentangled. Under these conditions, we provide a meta-algorithm called *Personalized Matching and Averaging (*PerMA*)* that can recover both global and local dictionaries from heterogeneous datasets. PerMA is highly efficient; it converges to the ground truth at a linear rate under suitable conditions. Moreover, it automatically borrows strength from strong learners to improve the prediction of weak learners. As a general framework for extracting global and local dictionaries, we show the application of PerDL in different learning tasks, such as training with imbalanced datasets and video surveillance.

## 1 Introduction

Given a set of $n$ signals $\mathbf{Y} = [\mathbf{y}_1, \ldots, \mathbf{y}_n] \in \mathbb{R}^{d \times n}$, *dictionary learning* (DL) aims to find a *dictionary* $\mathbf{D} \in \mathbb{R}^{d \times r}$ and a corresponding *code* $\mathbf{X} = [\mathbf{x}_1, \ldots, \mathbf{x}_n] \in \mathbb{R}^{r \times n}$ such that: (1) each data sample $\mathbf{y}_i$ can be written as $\mathbf{y}_i = \mathbf{D}\mathbf{x}_i$ for $1 \leq i \leq n$, and (2) the code $\mathbf{X}$ has as few nonzero elements as possible. The columns of the dictionary $\mathbf{D}$, also known as *atoms*, encode the "common features" whose linear combinations form the data samples. A typical approach to solve DL is via the following optimization problem:

$$\min_{\mathbf{X}, \mathbf{D}} \|\mathbf{Y} - \mathbf{D}\mathbf{X}\|_F^2 + \lambda \|\mathbf{X}\|_{\ell_q}, \tag{DL}$$

Here $\| \cdot \|_{\ell_q}$ is often modeled as a $\ell_1$-norm (Arora et al., 2015; Agarwal et al., 2016) or $\ell_0$-(pseudo-)norm (Spielman et al., 2012; Liang et al., 2022) and has the role of promoting sparsity in the estimated sparse code. Due to its effective feature extraction and representation, DL has found immense applications in data analytics, with applications ranging from clustering and classification (Ramirez et al. (2010); Tošić and Frossard (2011)), to image denoising (Li et al. (2011)), to document detection(Kasiviswanathan et al. (2012)), to medical imaging (Zhao et al. (2021)), and to many others.

37th Conference on Neural Information Processing Systems (NeurIPS 2023).

However, the existing formulations of DL hinge on a critical assumption: the homogeneity of the data. It is assumed that the samples share the same set of features (atoms) collected in a *single* dictionary $D$. This assumption, however, is challenged in practice as the data is typically collected and processed in heterogeneous edge devices (clients). These clients (for instance, smartphones and wearable devices) operate in different conditions (Kontar et al., 2017) while sharing some congruity. Accordingly, the collected datasets are naturally endowed with heterogeneous features while potentially sharing common ones. In such a setting, the classical formulation of DL faces a major dilemma: on the one hand, a reasonably-sized dictionary (with a moderate number of atoms $r$) may overlook the unique features specific to different clients. On the other hand, collecting both shared and unique features in a single enlarged dictionary (with a large number of atoms $r$) may lead to computational, privacy, and identifiability issues. In addition, both approaches fail to provide information about "what is shared and unique" which may offer standalone intrinsic value and can potentially be exploited for improved clustering, classification and anomaly detection, amongst others.

With the goal of addressing data heterogeneity in dictionary learning, in this paper, we propose *personalized dictionary learning* (PerDL); a framework that can untangle and recover *global* and *local (unique)* dictionaries from heterogeneous datasets. The global dictionary, which collects atoms that are shared among all clients, represents the common patterns among datasets and serves as a conduit of collaboration in our framework. The local dictionaries, on the other hand, provide the necessary flexibility for our model to accommodate data heterogeneity.

We summarize our contributions below:

- *Identifiability of local and global atoms:* We provide conditions under which the local and global dictionaries can be provably identified and separated by solving a nonconvex optimization problem. At a high level, our identifiability conditions entail that the true dictionaries are column-wise incoherent, and the local atoms do not have a significant alignment along any nonzero vector.

- *Federated meta-algorithm:* We present a fully federated meta-algorithm, called PerMA (Algorithm 1), for solving PerDL. PerMA only requires communicating the estimated dictionaries among the clients, thereby circumventing the need for sharing any raw data. A key property of PerMA is its ability to untangle global and local dictionaries by casting it as a series of shortest path problems over a *directed acyclic graph* (DAG).

- *Theoretical guarantees:* We prove that, under moderate conditions on the generative model and the clients, PerMA enjoys a linear convergence to the ground truth up to a statistical error. Additionally, PerMA borrows strength from strong clients to improve the performance of the weak ones. More concretely, through collaboration, our framework provides weak clients with the extra benefits of *averaged* initial condition, convergence rate, and final statistical error.

- *Practical performance:* We showcase the performance of PerMA on a synthetic dataset, as well as different realistic learning tasks, such as training with imbalanced datasets and video surveillance. These experiments highlight that our method can effectively extract shared global features while preserving unique local ones, ultimately improving performance through collaboration.

## 1.1 Related Works

**Dictionary Learning**   Spielman et al. (2012); Liang et al. (2022) provide conditions under which DL can be provably solved, provided that the dictionary is a square matrix (also known as *complete* DL). For the more complex case of *overcomplete* DL with $r > d$, Arora et al. (2014, 2015); Agarwal et al. (2016) show that alternating minimization achieves desirable statistical and convergence guarantees. Inspired by recent results on the benign landscape of matrix factorization (Ge et al., 2017; Fattahi and Sojoudi, 2020), Sun et al. (2016) show that a smoothed variant of DL is devoid of spurious local solutions. In contrast, distributed or federated variants of DL are far less explored. Huang et al. (2022); Gkillas et al. (2022) study DL in the federated setting. However, they do not provide any provable guarantees on their proposed method.

**Federated Learning & Personalization**   Recent years have seen explosive interest in federated learning (FL) following the seminal paper on federated averaging (McMahan et al., 2017). Literature

along this line has primarily focused on predictive modeling using deep neural networks (DNN), be it through enabling faster convergence (Karimireddy et al., 2020), improving aggregation schemes at a central server (Wang et al., 2020), promoting fairness across all clients (Yue et al., 2022) or protecting against potential adversaries (Bhagoji et al., 2019). A comprehensive survey of existing methodology can be found in (Kontar et al., 2021). More recently, the focus has been placed on tackling heterogeneity across client datasets through personalization. The key idea is to allow each client to retain their own tailored model instead of learning one model that fits everyone. Approaches along this line either split weights of a DNN into shared and unique ones and collaborate to learn the shared weights (Liang et al., 2020; T Dinh et al., 2020), or follow a train-then-personalize approach where a global model is learned and fine-tuned locally, often iteratively (Li et al., 2021). Again such models have mainly focused on predictive models. Whilst this literature abounds, personalization that aims to identify what is shared and unique across datasets is very limited. Very recently, personalized PCA (Shi and Kontar, 2022) was proposed to address this challenge through identifiably extracting shared and unique principal components using distributed Riemannian gradient descent. However, PCA cannot accommodate sparsity in representation and requires orthogonality constraints that may limit its application. In contrast, our work considers a broader setting via sparse dictionary learning.

**Notation.** For a matrix $\mathbf{A}$, we use $\|\mathbf{A}\|_2$, $\|\mathbf{A}\|_F$, $\|\mathbf{A}\|_{1,2}$, and $\|\mathbf{A}\|_1$ to denote its spectral norm, Frobenius norm, the maximum of its column-wise 2-norm, and the element-wise 1-norm of $\mathbf{A}$, respectively. We use $\mathbf{A}_i$ to indicate that it belongs to client $i$. Moreover, we use $\mathbf{A}_{(i)}$ to denote the $i$-th column of $\mathbf{A}$. We use $\mathcal{P}(n)$ to denote the set of $n \times n$ signed permutation matrices. We define $[n] = \{1, 2, \ldots, n\}$.

## 2 PerDL: **Personalized Dictionary Learning**

In PerDL, we are given $N$ clients, each with $n_i$ samples collected in $\mathbf{Y}_i \in \mathbb{R}^{d \times n_i}$ and generated as a sparse linear combination of $r^g$ global atoms and $r_i^l$ local atoms:

$$\mathbf{Y}_i = \mathbf{D}_i^* \mathbf{X}_i^*, \quad \text{where} \quad \mathbf{D}_i^* = \begin{bmatrix} \mathbf{D}^{g*} & \mathbf{D}_i^{l*} \end{bmatrix}, \quad \text{for} \quad 1 = 1, \ldots, N. \tag{1}$$

Here $\mathbf{D}^{g*} \in \mathbb{R}^{d \times r^g}$ denotes a global dictionary that captures the common features shared among all clients, whereas $\mathbf{D}_i^{l*} \in \mathbb{R}^{d \times r_i^l}$ denotes the local dictionary specific to each client. Let $r_i = r^g + r_i^l$ denote the total number of atoms in $\mathbf{D}_i^*$. Without loss of generality, we assume the columns of $\mathbf{D}_i^*$ have unit $\ell_2$-norm.[1] The goal in PerDL is to recover $\mathbf{D}^{g*}$ and $\{\mathbf{D}_i^{l*}\}_{i=1}^N$, as well as the sparse codes $\{\mathbf{X}_i^*\}_{i=1}^N$, given the datasets $\{\mathbf{Y}_i\}_{i=1}^N$. Before presenting our approach for solving PerDL, we first consider the following fundamental question: under what conditions is the recovery of the dictionaries $\mathbf{D}^{g*}$, $\{\mathbf{D}_i^{l*}\}_{i=1}^N$ and sparse codes $\{\mathbf{X}_i^*\}_{i=1}^N$ well-posed?

To answer this question, we first note that it is only possible to recover the dictionaries and sparse codes up to a signed permutation: given any signed permutation matrix $\mathbf{\Pi} \in \mathcal{P}(r_i)$, the dictionary-code pairs $(\mathbf{D}_i, \mathbf{X}_i)$ and $(\mathbf{D}_i \mathbf{\Pi}_i, \mathbf{\Pi}_i^\top \mathbf{X}_i)$ are equivalent. This invariance with respect to signed permutation gives rise to an equivalent class of true solutions with a size that grows exponentially with the dimension. To guarantee the recovery of a solution from this equivalent class, we need the $\mu$-incoherency of the true dictionaries.

**Assumption 1** ($\mu$-incoherency). *For each client $1 \leq i \leq N$, the dictionary $\mathbf{D}_i^*$ is $\mu$-incoherent for some constant $\mu > 0$, that is,*

$$\max_{j,k} \left| \left\langle (\mathbf{D}_i^*)_{(j)}, (\mathbf{D}_i^*)_{(k)} \right\rangle \right| \leq \frac{\mu}{\sqrt{d}}. \tag{2}$$

Assumption 1 is standard in dictionary learning (Agarwal et al. (2016); Arora et al. (2015); Chatterji and Bartlett (2017)) and was independently introduced by Fuchs (2005); Tropp (2006) in signal processing and Zhao and Yu (2006); Meinshausen and Bühlmann (2006) in statistics. Intuitively, it requires the atoms in each dictionary to be approximately orthogonal. To see the necessity of this assumption, consider a scenario where two atoms of $\mathbf{D}_i^*$ are perfectly aligned (i.e., $\mu = \sqrt{d}$). In this case, using either of these two atoms achieve a similar effect in reconstructing $\mathbf{Y}_i$, contributing to the ill-posedness of the problem.

---

[1]This assumption is without loss of generality since, for any dictionary-code pair $(\mathbf{D}_i, \mathbf{X}_i)$, the columns of $\mathbf{D}_i$ can be normalized to have unit norm by re-weighting the corresponding rows of $\mathbf{X}_i$.

Our next assumption guarantees the separation of local dictionaries from the global one in PerDL. First, we introduce several signed permutation-invariant distance metrics for dictionaries, which will be useful for later analysis.

**Definition 1.** *For two dictionaries $\mathbf{D}_1, \mathbf{D}_2 \in \mathbb{R}^{d \times r}$, we define their signed permutation-invariant $\ell_{1,2}$-distance and $\ell_2$-distance as follows:*

$$d_{1,2}(\mathbf{D}_1, \mathbf{D}_2) := \min_{\mathbf{\Pi} \in \mathcal{P}(r)} \|\mathbf{D}_1 \mathbf{\Pi} - \mathbf{D}_2\|_{1,2}, \tag{3}$$

$$d_2(\mathbf{D}_1, \mathbf{D}_2) := \min_{\mathbf{\Pi} \in \mathcal{P}(r)} \|\mathbf{D}_1 \mathbf{\Pi} - \mathbf{D}_2\|_2. \tag{4}$$

*Furthermore, suppose $\mathbf{\Pi}^* = \arg\min_{\mathbf{\Pi} \in \mathcal{P}(r)} \|\mathbf{D}_1 \mathbf{\Pi} - \mathbf{D}_2\|_{1,2}$. For any $1 \leq j \leq r$, we define*

$$d_{2,(j)}(\mathbf{D}_1, \mathbf{D}_2) := \left\| (\mathbf{D}_1 \mathbf{\Pi}^* - \mathbf{D}_2)_{(j)} \right\|_2. \tag{5}$$

**Assumption 2** ($\beta$-identifiablity)**.** *The local dictionaries $\left\{ \mathbf{D}_i^{l*} \right\}_{i=1}^N$ are $\beta$-identifiable for some constant $0 < \beta < 1$, that is, there exists no vector $\mathbf{v} \in \mathbb{R}^d$ with $\|\mathbf{v}\|_2 = 1$ such that*

$$\max_{1 \leq i \leq N} \min_{1 \leq j \leq r_l} d_2 \left( \left( \mathbf{D}_i^{l*} \right)_{(j)}, \mathbf{v} \right) \leq \beta. \tag{6}$$

Suppose there exists a unit-norm $\mathbf{v}$ satisfying (6) for some small $\beta > 0$. This implies that $\mathbf{v}$ is sufficiently close to at least one atom from each local dictionary. Indeed, one may treat this atom as part of the global dictionary, thereby violating the identifiability of local and global dictionaries. On the other hand, the infeasibility of (6) for large $\beta > 0$ implies that the local dictionaries are sufficiently dispersed, which in turn facilitates their identification.

With the above assumptions in place, we are ready to present our proposed optimization problem for solving PerDL:

$$\min_{\mathbf{D}^g, \{\mathbf{D}_i\}, \{\mathbf{X}_i\}} \sum_{i=1}^N \|\mathbf{Y}_i - \mathbf{D}_i \mathbf{X}_i\|_F^2 + \lambda \sum_{i=1}^N \|\mathbf{X}_i\|_{\ell_q}, \quad \text{s.t.} \quad (\mathbf{D}_i)_{(1:r^g)} = \mathbf{D}^g \quad \text{for} \quad 1 \leq i \leq N.$$
$$\text{(PerDL-NCVX)}$$

For each client $i$, `PerDL-NCVX` aims to recover a dictionary-code pair $(\mathbf{D}_i, \mathbf{X}_i)$ that match $\mathbf{Y}_i$ under the constraint that dictionaries for individual clients share the same global components.

# 3   Meta-algorithm of Solving PerDL

In this section, we introduce our meta-algorithm (Algorithm 1) for solving `PerDL-NCVX`, which we call *Personalized Matching and Averaging* (PerMA). In what follows, we explain the steps of PerMA:

**Local initialization (Step 3):**   PerMA starts with a warm-start step where each client runs their own initialization scheme to obtain $\mathbf{D}_i^{(0)}$. This step is necessary even for the classical DL to put the initial point inside a basin of attraction of the ground truth. Several spectral methods were proposed to provide a theoretically good initialization(Arora et al., 2015; Agarwal et al., 2016), while in practice, it is reported that random initialization followed by a few iterations of alternating minimization approach will suffice (Ravishankar et al., 2020; Liang et al., 2022).

**Global matching scheme (Step 6)**   Given the clients' initial dictionaries, our global matching scheme separates the global and local parts of each dictionary by solving a series of shortest path problems on an auxiliary graph. Then, it obtains a refined estimate of the global dictionary via simple averaging. A detailed explanation of this step is provided in the next section.

**Dictionary update at each client (Step 10)**   During each communication round, the clients refine their own dictionary based on the available data, the aggregated global dictionary, and the previous estimate of their local dictionary. A detailed explanation of this step is provided in the next section.

**Algorithm 1** PerMA: Federated Matching and Averaging

1: **Input:** $\{\mathbf{Y}_i\}_{i=1}^N$.
2: **for** client $i = 1, ..., N$ **do**
3:    *Client:* Obtain $\mathbf{D}_i^{(0)}$ based on $\mathbf{Y}_i$.                  `// Initialization step`
4:    *Client:* Send $\mathbf{D}_i^{(0)}$ to the server.
5: **end for**
6: *Server:* $\left(\mathbf{D}^{g,(0)}, \{\mathbf{D}_i^{l,(0)}\}_{i=1}^N\right) = \texttt{global\_matching}\left(\{\mathbf{D}_i^{(0)}\}_{i=1}^N\right)$
                                 `// Separating local from global dictionaries`
7: *Server:* Broadcast $\left(\mathbf{D}^{g,(0)}, \{\mathbf{D}_i^{l,(0)}\}_{i=1}^N\right)$
8: **for** $t = 0, 1, \ldots, T$ **do**
9:    **for** client $i = 1, ..., N$ **do**
10:       *Client:* $\left(\mathbf{D}_i^{g,(t+1)}, \mathbf{D}_i^{l,(t+1)}\right) = \texttt{local\_update}\left(\mathbf{Y}_i, \mathbf{D}^{g,(t)}, \mathbf{D}_i^{l,(t)}\right)$
                      `// Updating the local and global dictionaries for each client`
11:       *Client:* Send $\mathbf{D}_i^{g,(t+1)}$ to the server.
12:    **end for**
13:    *Server:* Calculate $\mathbf{D}^{g,(t+1)} = \frac{1}{N}\sum_{i=1}^N \mathbf{D}_i^{g,(t+1)}$. `// Averaging global dictionaries`
14:    *Server:* Broadcast $\mathbf{D}^{g,(t+1)}$.
15: **end for**
16: **return** $\left(\mathbf{D}^{g,(T)}, \{\mathbf{D}_i^{l,(T)}\}_{i=1}^N\right)$.

**Global aggregation (Step 13)** At the end of each round, the server updates the clients' estimate of the global dictionary by computing their average.

A distinguishing property of PerMA is that it only requires the clients to communicate their dictionaries and not their sparse codes. In fact, after the global matching step on the initial dictionaries, the clients only need to communicate their global dictionaries, keeping their local dictionaries private.

### 3.1 Global Matching and Local Updates

In this section, we provide detailed implementations of `global_matching` and `local_update` subroutines in PerMA (Algorithm 1).

Given the initial approximations of the clients' dictionaries $\{\mathbf{D}_i^{(0)}\}_{i=1}^N$, `global_matching` seeks to identify and aggregate the global dictionary by extracting the similarities among the atoms of $\{\mathbf{D}_i^{(0)}\}$. To identify the global components, one approach is to solve the following optimization problem

$$\min_{\mathbf{\Pi}_i} \sum_{i=1}^{N-1} \left\|\left(\mathbf{D}_i^{(0)}\mathbf{\Pi}_i\right)_{(1:r^g)} - \left(\mathbf{D}_{i+1}^{(0)}\mathbf{\Pi}_{i+1}\right)_{(1:r^g)}\right\|_2 \quad \text{s.t.} \quad \mathbf{\Pi}_i \in \mathcal{P}(r_i) \quad \text{for} \quad 1 \le i \le N. \tag{7}$$

The above optimization aims to obtain the appropriate signed permutation matrices $\{\mathbf{\Pi}_i\}_{i=1}^N$ that align the first $r^g$ atoms of the permuted dictionaries. In the ideal regime where $\mathbf{D}_i^{(0)} = \mathbf{D}_i^*, 1 \le i \le N$, the optimal solution $\{\mathbf{\Pi}_i^*\}_{i=1}^N$ yields a zero objective value and satisfies $\left(\mathbf{D}_i^{(0)}\mathbf{\Pi}_i^*\right)_{(1:r^g)} = \mathbf{D}^{g*}$. However, there are two main challenges with the above optimization. First, it is a nonconvex, combinatorial problem over the discrete sets $\{\mathcal{P}(r_i)\}$. Second, the initial dictionaries may not

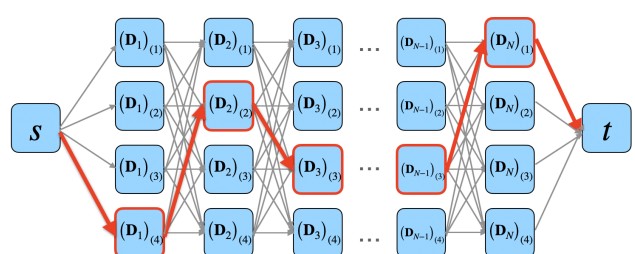

Figure 1: A schematic diagram for `global_matching` (Algorithm 2). At each iteration, we find the shortest path from $s$ to $t$ (highlighted with red), estimate one atom of $\mathbf{D}^{g*}$ using all passed vertices and remove the path (including the vertices) from $\mathcal{G}$.

coincide with their true counterparts. To address the first challenge, we show that the optimal solution to the optimization (7) can be efficiently obtained by solving a series of shortest path problems defined over an auxiliary graph. To alleviate the second challenge, we show that our proposed algorithm is robust against possible errors in the initial dictionaries.

Consider a weighted $N$-layered *directed acyclic graph* (DAG) $\mathcal{G}$ with $r_i$ nodes in layer $i$ representing the $r_i$ atoms in $\mathbf{D}_i$. We connect any node $a$ from layer $i$ to any node $b$ from layer $i+1$ with a directed edge with weight $w(a, b) = d_2\left((\mathbf{D}_i)_a, (\mathbf{D}_{i+1})_b\right)$. We add a *source* node $s$ and connect it to all nodes in layer 1 with weight 0. Similarly, we include a *terminal* node $t$ and connect all nodes in layer $N$ to $t$ with weight 0. A schematic construction of this graph is presented in Figure 1. Given the constructed graph, Algorithm 2 aims to solve (7) by running $r^g$ rounds of the shortest path problem: at each round, the algorithm identifies the most aligned atoms in the initial dictionaries by obtaining the shortest path from $s$ to $t$. Then it removes the used nodes in the path for the next round. The correctness and robustness of the proposed algorithm are established in the next theorem.

---

**Algorithm 2** `global_matching`

---

1: **Input:** $\left\{\mathbf{D}_i^{(0)}\right\}_{i=1}^{N}$ and $r^g$.
2: Construct the weighted $N$-layer DAG $\mathcal{G}$ described in Section 3.1.
3: Initialize $\{\text{index}_i\}_{i=1}^{N}$ as empty sets and $\mathbf{D}^{g,(0)}$ as an empty matrix.
4: **for** $j = 1, \ldots, r^g$ **do**
5:     Find the shortest path $\mathcal{P} = \left(s, (\mathbf{D}_1^{(0)})_{(\alpha_1)}, (\mathbf{D}_1^{(0)})_{(\alpha_2)}, \cdots, (\mathbf{D}_N^{(0)})_{(\alpha_N)}, t\right)$.
6:     Add $\frac{1}{N}\sum_{i=1}^{N} \text{sign}\left(\left\langle (\mathbf{D}_i^{(0)})_{(\alpha_i)}, (\mathbf{D}_1^{(0)})_{(\alpha_1)}\right\rangle\right)(\mathbf{D}_i^{(0)})_{(\alpha_i)}$ as a new column of $\mathbf{D}^{g,(0)}$.
7:     Add $\alpha_i$ to $\text{index}_i$ for every $i = 1, \ldots, N$.
8:     Remove $\mathcal{P}$ from $\mathcal{G}$.
9: **end for**
10: Set $\mathbf{D}_i^{l,(0)} = (\mathbf{D}_i^{(0)})_{([r_i]\setminus\text{index}_i)}$ for every $i = 1, \ldots, N$.
11: **return** $\left(\mathbf{D}^{g,(0)}, \left\{\mathbf{D}_i^{l,(0)}\right\}_{i=1}^{N}\right)$.

---

**Theorem 1** (Correctness and robustness of `global_matching`). *Suppose $\{\mathbf{D}_i^*\}_{i=1}^{N}$ are $\mu$-incoherent (Assumption 1) and $\beta$-identifiable (Assumption 2). Suppose the initial dictionaries $\left\{\mathbf{D}_i^{(0)}\right\}_{i=1}^{N}$ satisfy $d_{1,2}\left(\mathbf{D}_i^{(0)}, \mathbf{D}_i^*\right) \leq \epsilon_i$ with $4\sum_{i=1}^{N}\epsilon_i \leq \min\left\{\sqrt{2 - 2\frac{\mu}{\sqrt{d}}}, \beta\right\}$. Then, the output of Algorithm 2 satisfies:*

$$d_{1,2}\left(\mathbf{D}^{g,(0)}, \mathbf{D}^{g*}\right) \leq \frac{1}{N}\sum_{i=1}^{N}\epsilon_i, \qquad and \qquad d_{1,2}\left(\mathbf{D}_i^{l,(0)}, \mathbf{D}_i^{l*}\right) \leq \epsilon_i, \quad for \quad 1 \leq i \leq N. \quad (8)$$

According to the above theorem, `global_matching` can robustly separate the clients' initial dictionaries into global and local parts, provided that the aggregated error in the initial dictionaries is below a threshold. Specific initialization schemes that can satisfy the condition of Theorem 1 include Algorithm 1 from Agarwal et al. (2013) and Algorithm 3 from Arora et al. (2015). We also remark that since the constructed graph is a DAG, the shortest path problem can be solved in time linear in the number of edges, which is $\mathcal{O}\left(r_1 + r_N + \sum_{i=1}^{N-1} r_i r_{i+1}\right)$, via a simple labeling algorithm (see, e.g., (Ahuja et al., 1988, Chapter 4.4)). Since we need to solve the shortest path problem $r^g$ times, this brings the computational complexity of Algorithm 2 to $\mathcal{O}(r^g N r_{\max}^2)$, where $r_{\max} = \max_i r_i$.

Given the initial local and global dictionaries, the clients progressively refine their estimates by applying $T$ rounds of `local_update` (Algorithm 3). At a high level, each client runs a single iteration of a *linearly convergent algorithm* $\mathcal{A}_i$ (see Definition 2), followed by an alignment step that determines the global atoms of the updated dictionary using $\mathbf{D}^{g,(t)}$ as a "reference point". Notably, our implementation of `local_update` is adaptive to different DL algorithms. This flexibility is indeed intentional to provide a versatile meta-algorithm for clients with different DL algorithms.

**Algorithm 3** `local_update`

1: **Input:** $\mathbf{D}_i^{(t)} = \begin{bmatrix} \mathbf{D}^{g,(t)} & \mathbf{D}_i^{l,(t)} \end{bmatrix}, \mathbf{Y}_i$

2: $\mathbf{D}_i^{(t+1)} = \mathcal{A}_i \left( \mathbf{Y}_i, \mathbf{D}_i^{(t)} \right)$ `// One iteration of a linearly-convergent algorithm.`

3: Initialize $\mathcal{S}$ as an empty set and $\mathbf{P} \in \mathbb{R}^{r^g \times r^g}$ as an all-zero matrix.

4: **for** $j = 1, ..., r^g$ **do**

5:     Find $k^* = \arg\min_k d_2 \left( \left( \mathbf{D}^{g,(t)} \right)_{(j)}, \left( \mathbf{D}_i^{(t+1)} \right)_{(k)} \right)$.

6:     Append $k^*$ to $\mathcal{S}$.

7:     Set $(i,i)$-th entry of $\mathbf{P}$ to sign $\left( \left\langle \left( \mathbf{D}^{g,(t)} \right)_{(j)}, \left( \mathbf{D}_i^{(t+1)} \right)_{(k^*)} \right\rangle \right)$.

8: **end for**

9: **Output:** $\mathbf{D}_i^{g,(t+1)} = \left( \mathbf{D}_i^{(t+1)} \right)_{(\mathcal{S})} \mathbf{P}$ and $\mathbf{D}_i^{l,(t+1)} = \left( \mathbf{D}_i^{(t+1)} \right)_{([r_i] \backslash \mathcal{S})}$.

## 4 Theoretical Guarantees

In this section, we show that our proposed meta-algorithm provably solves PerDL under suitable initialization, identifiability, and algorithmic conditions. To achieve this goal, we first present the definition of a linearly-convergent DL algorithm.

**Definition 2.** *Given a generative model $\mathbf{Y} = \mathbf{D}^* \mathbf{X}^*$, a DL algorithm $\mathcal{A}$ is called $(\delta, \rho, \psi)$-linearly convergent for some parameters $\delta, \psi > 0$ and $0 < \rho < 1$ if, for any $\mathbf{D} \in \mathbb{R}^{d \times r}$ such that $d_{1,2}(\mathbf{D}, \mathbf{D}^*) \leq \delta$, the output of one iteration $\mathbf{D}^+ = \mathcal{A}(\mathbf{D}, \mathbf{Y})$, satisfies*

$$d_{2,(j)} \left( \mathbf{D}^+, \mathbf{D}^* \right) \leq \rho d_{2,(j)} \left( \mathbf{D}, \mathbf{D}^* \right) + \psi, \quad \forall 1 \leq j \leq r. \tag{9}$$

One notable linearly convergent algorithm is introduced by (Arora et al., 2015, Algorithm 5); we will discuss this algorithm in more detail in the appendix. Assuming all clients are equipped with linearly convergent algorithms, our next theorem establishes the convergence of PerMA.

**Theorem 2** (Convergence of PerMA). *Suppose $\{\mathbf{D}_i^*\}_{i=1}^N$ are $\mu$-incoherent (Assumption 1) and $\beta$-identifiable (Assumption 2). Suppose, for every client $i$, the DL algorithm $\mathcal{A}_i$ used in* `local_update` *(Algorithm 3) is $(\delta_i, \rho_i, \psi_i)$-linearly convergent with $4 \sum_{i=1}^N \delta_i \leq \min \left\{ \sqrt{2 - 2\frac{\mu}{\sqrt{d}}}, \beta \right\}$. Assume the initial dictionaries $\{\mathbf{D}_i^{(0)}\}_{i=1}^N$ satisfy:*

$$d_{1,2} \left( \frac{1}{N} \sum_{i=1}^N \mathbf{D}_i^{g,(0)}, \mathbf{D}^{g*} \right) \leq \min_{1 \leq i \leq N} \delta_i, \quad d_{1,2} \left( \mathbf{D}_i^{l,(0)}, \mathbf{D}_i^{l*} \right) \leq \delta_i, \quad for \quad i = 1, \dots, N. \tag{10}$$

*Then, for every $t \geq 0$, PerMA (Algorithm 1) satisfies*

$$d_{1,2} \left( \mathbf{D}^{g,(t)}, \mathbf{D}^{g*} \right) \leq \left( \frac{1}{N} \sum_{i=1}^N \rho_i \right) d_{1,2} \left( \mathbf{D}^{g,(0)}, \mathbf{D}^{g*} \right) + \frac{1}{N} \sum_{i=1}^N \psi_i, \tag{11}$$

$$d_{1,2} \left( \mathbf{D}_i^{l,(t)}, \mathbf{D}_i^{l*} \right) \leq \rho_i d_{1,2} \left( \mathbf{D}_i^{l,(0)}, \mathbf{D}_i^{l*} \right) + \psi_i, \quad for \quad 1 \leq i \leq N. \tag{12}$$

The above theorem sheds light on a number of key benefits of PerMA:

**Relaxed initial condition for weak clients.** Our algorithm relaxes the initial condition on the global dictionaries. In particular, it only requires the average of the initial global dictionaries to be close to the true global dictionary. Consequently, it enjoys a provable convergence guarantee even if some of the clients do not provide a high-quality initial dictionary to the server.

**Improved convergence rate for slow clients.** During the course of the algorithm, the global dictionary error decreases at an average rate of $\frac{1}{N} \sum_{i=1}^N \rho_i$, improving upon the convergence rate of the slow clients.

**Improved statistical error for weak clients.** A linearly convergent DL algorithm $\mathcal{A}_i$ stops making progress upon reaching a neighborhood around the true dictionary $\mathbf{D}_i^*$ with radius $O(\psi_i)$. This type of guarantee is common among DL algorithms (Arora et al., 2015; Liang et al., 2022) and often corresponds to their statistical error. In light of this, PerMA improves the performance of weak clients (i.e., clients with weak statistical guarantees) by borrowing strength from strong ones.

## 5 Numerical Experiments

In this section, we showcase the effectiveness of Algorithm 1 using synthetic and real data. All experiments are performed on a MacBook Pro 2021 with the Apple M1 Pro chip and 16GB unified memory for a serial implementation in MATLAB 2022a. Due to limited space, we will only provide the high-level motivation and implication of our experiments. We defer implementation details and comparisons with the existing methods to the appendix.

### 5.1 Synthetic Dataset

In this section, we validate our theoretical results on a synthetic dataset. We consider ten clients, each with a dataset generated according to the model 1. The details of our construction are presented in the appendix. Specifically, we compare the performances of two strategies: (1) *independent strategy*, where each client solves DL without any collaboration, and (2) *collaborative strategy*, where clients collaboratively learn the ground truth dictionaries by solving PerDL via the proposed meta-algorithm PerMA. We initialize both strategies using the same $\{\mathbf{D}_i^{(0)}\}_{i=1}^N$. The initial dictionaries are obtained via a warm-up method proposed in (Liang et al., 2022, Algorithm 4). For a fair comparison between independent and collaborative strategies, we use the same DL algorithm ((Liang et al., 2022, Algorithm 1)) for different clients. Note that in the independent strategy, the clients cannot separate global

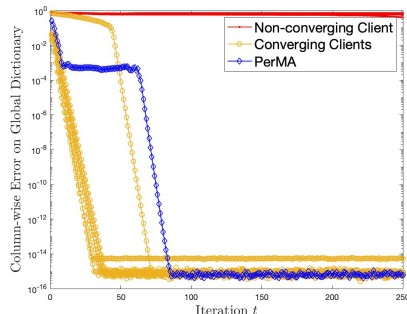

Figure 2: PerMA improves the accuracy of the recovered global dictionary for *all* clients, even if some (three out of ten) are weak learners.

from local dictionaries. Nonetheless, to evaluate their performance, we collect the atoms that best align with the true global dictionary $\mathbf{D}^{g*}$ and treat them as the estimated global dictionaries. As can be seen in Figure 2, three out of ten clients are weak learners and fail to recover the global dictionary with desirable accuracy. On the contrary, in the collaborative strategy, all clients recover the same global dictionary almost exactly.

### 5.2 Training with Imbalanced Data

In this section, we showcase the application of PerDL in training with imbalanced datasets. We consider an image reconstruction task on MNIST dataset. This dataset corresponds to a set of handwritten digits (see the first row of Figure 3). The goal is to recover a *single* concise global dictionary that can be used to reconstruct the original handwritten digits as accurately as possible. In particular, we study a setting where the clients have *imbalanced label distributions*. Indeed, data imbalance can drastically bias the performance of the trained model in favor of the majority groups, while hurting its performance on the minority groups (Leevy et al., 2018; Thabtah et al., 2020). Here, we consider a setting where the clients have highly imbalanced datasets, where 90% of their samples have the same label. More specifically, for client $i$, we assume that 90% of the samples correspond to the handwritten digit "$i$", with the remaining 10% corresponding to other digits. The second row of Figure 3 shows the effect of data imbalance on the performance of the recovered dictionary on a single client, when the clients do not collaborate. The last row of Figure 3 shows the improved performance of the recovered dictionary via PerDL on the same client. Our experiment clearly shows the ability of PerDL to effectively address the data imbalance issue by combining the strengths of different clients.

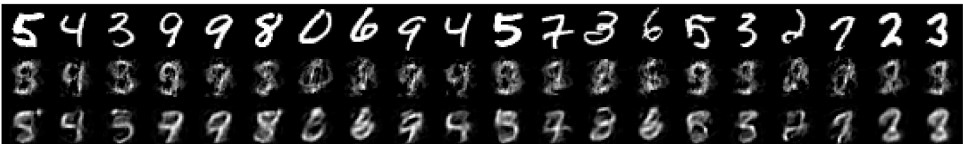

Figure 3: PerMA improves training with imbalanced datasets. We consider the image reconstruction task on the imbalanced MNIST dataset using only five atoms from a learned global dictionary. The first row corresponds to the original images. The second row is based on the dictionary learned on a single client with an imbalanced dataset. The third row shows the improved performance of the learned dictionary using our proposed method on the same client.

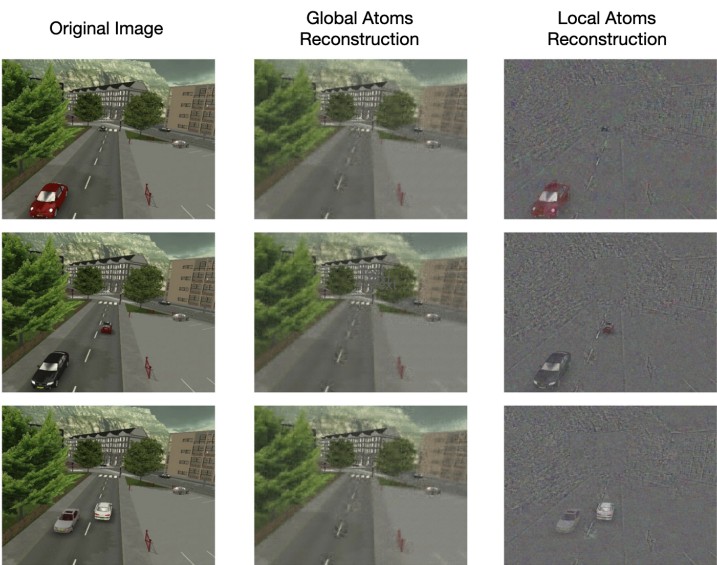

Figure 4: PerMA effectively separates the background from moving objects in video frames. Here we reconstruct the surveillance video frames using global and local dictionaries learned by PerMA. We reconstruct the frames using only $50$ atoms from the combined dictionaries.

## 5.3 Surveillance Video Dataset

As a proof of concept, we consider a video surveillance task, where the goal is to separate the background from moving objects. Our data is collected from Vacavant et al. (2013) (see the first column of Figure 4). As these frames are taken from one surveillance camera, they share the same background corresponding to the global features we aim to extract. The frames also exhibit heterogeneity as moving objects therein are different from the background. This problem can indeed be modeled as an instance of PerDL, where each video frame can be assigned to a "client", with the global dictionary capturing the background and local dictionaries modeling the moving objects. We solve PerDL by applying PerMA to obtain a global dictionary and several local dictionaries for this dataset. Figure 4 shows the reconstructed background and moving objects via the recovered global and local dictionaries. Our results clearly show the ability of our proposed framework to separate global and local features. [2]

---

[2]We note that moving object detection in video frames has been extensively studied in the literature and typically solved very accurately via different representation learning methods (such as robust PCA and neural network modeling); see (Yazdi and Bouwmans, 2018) for a recent survey. Here, we use this case study as a proof of concept to illustrate the versatility of PerMA in handling heterogeneity, even in settings where the data is not physically distributed among clients.

# 6 Social Impact, Limitations and Future Directions

Our novel approach for personalized dictionary learning presents a versatile solution with immediate applications across various domains, such as video surveillance and object detection. While these applications offer valuable benefits, they also bring to the forefront ethical and societal concerns, particularly concerning privacy, bias, and potential misuse. In the context of video surveillance, deploying object detection algorithms may inadvertently capture private information, leading to concerns about violating individuals' right to privacy. However, it is important to emphasize that our proposed algorithm is specifically designed to focus on separating unique and common features within the data, without delving into the realm of personal information. Thus, it adheres to ethical principles by ensuring that private data is not processed or used without explicit consent. Bias is another critical aspect that necessitates careful consideration in the deployment of object detection algorithms. Biases can manifest in various forms, such as underrepresentation or misclassification of certain groups, leading to discriminatory outcomes. Our approach acknowledges the importance of mitigating biases by solely focusing on the distinction between common and unique features, rather than introducing any inherent bias into the learning process. Furthermore, the potential misuse of object detection algorithms in unauthorized surveillance or invasive tracking scenarios raises valid concerns. As responsible researchers, we are cognizant of such risks and stress that our proposed algorithm is meant to be deployed in a controlled and legitimate manner, adhering to appropriate regulations and ethical guidelines.

Even though our meta-algorithm PerMA enjoys strong theoretical guarantees and practical performance, there are still several avenues for improvement. For instance, the theoretical success of PerMA, especially the Global Matching step, relies on an individual initial error of $O(1/N)$. In other words, the initial error should decrease as the number of clients grows. As a future work, we plan to relax such dependency via a more delicate analysis. We also note that imposing an upper bound on the initial error is not unique to our setting, as virtually all existing algorithms for classical (non-personalized) dictionary learning require certain conditions on the initial error. On the other hand, once the assumption on the initial error is satisfied, our meta-algorithm achieves a final error with the same dependency on $d$ (the dimensionality of the data) and $n$ (the number of samples) as the state-of-the-art algorithms for classical dictionary learning (Agarwal et al. (2016), Arora et al. (2015)). Remarkably, this implies that personalization is achieved without incurring any additional cost on $d$ or $n$, making PerMA highly efficient and competitive in its performance.

## Acknowledgements

S.F. is supported, in part, by NSF Award DMS-2152776, ONR Award N00014-22-1-2127, and MICDE Catalyst Grant. R. A. K. is supported in part by NSF CAREER 2144147.

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

# A  Further Details on the Experiments

In this section, we provide further details on the numerical experiments reported in Section 5.

## A.1  Details of Section 5.1

In Section 5.1, we generate the synthetic datasets according to Model 1 with $N = 10$, $d = 6$, $r_i = 6$, and $n_i = 200$ for each $1 \leq i \leq 10$. Each $\mathbf{D}_i^*$ is an orthogonal matrix with the first $r^g = 3$ columns shared with every other client and the last $r_i^l = 3$ columns unique to themselves. Each $\mathbf{X}_i^*$ is first generated from a Gaussian-Bernoulli distribution where each entry is non-zero with a probability 0.2. Then, $\mathbf{X}_i^*$ is further truncated, where all the entries $(\mathbf{X}_i^*)_{(j,k)}$ with $|(\mathbf{X}_i^*)_{(j,k)}| < 0.3$ are replaced by $(\mathbf{X}_i^*)_{(j,k)} = 0.3 \times \text{sign}((\mathbf{X}_i^*)_{(j,k)})$.

We use the orthogonal DL algorithm (Algorithm 4) introduced in (Liang et al., 2022, Algorithm 1) as the local DL algorithm for each client. This algorithm is simple to implement and comes equipped with a strong convergence guarantee (see (Liang et al., 2022, Theorem 1)). Here $\text{HT}_\zeta(\cdot)$ denotes the hard-thresholding operator at level $\zeta$, which is defined as:

$$(\text{HT}_\zeta(\mathbf{A}))_{(i,j)} = \begin{cases} \mathbf{A}_{(i,j)} & \text{if} \quad |\mathbf{A}_{(i,j)}| \geq \zeta, \\ 0 & \text{if} \quad |\mathbf{A}_{(i,j)}| < \zeta. \end{cases}$$

Specifically, we use $\zeta = 0.15$ for the experiments in Section 5.1. $\text{Polar}(\cdot)$ denotes the polar decomposition operater, which is defined as $\text{Polar}(\mathbf{A}) = \mathbf{U_A} \mathbf{V_A}^\top$, where $\mathbf{U_A} \mathbf{\Sigma_A} \mathbf{V_A}^\top$ is the Singular Value Decomposition (SVD) of $\mathbf{A}$.

---

**Algorithm 4** Alternating minimization for orthogonal dictionary learning (Liang et al. (2022))

---

1: **Input:** $\mathbf{Y}_i, \mathbf{D}_i^{(t)}$
2: Set $\mathbf{X}_i^{(t)} = \text{HT}_\zeta \left( \mathbf{D}^{(t)_i \top} \mathbf{Y}_i \right)$
3: Set $\mathbf{D}_i^{(t+1)} = \text{Polar} \left( \mathbf{Y}_i \mathbf{X}_i^{(t)\top} \right)$
4: **return** $\mathbf{D}_i^{(t+1)}$

---

For a fair comparison, we initialize both strategies using the same $\{\mathbf{D}_i^{(0)}\}_{i=1}^N$, which is obtained by iteratively calling Algorithm 4 with a random initial dictionary and shrinking thresholds $\zeta$. For a detailed discussion on such an initialization scheme we refer the reader to Liang et al. (2022).

## A.2  Details of Section 5.2

In section 5.2, we aim to learn a dictionary with imbalanced data collected from MNIST dataset (LeCun et al., 2010). Specifically, we consider $N = 10$ clients, each with 500 handwritten images. Each image is comprised of $28 \times 28$ pixels. Instead of randomly assigning images, we construct dataset $i$ such that it contains 450 images of digit $i$ and 50 images of other digits. Here client 10 corresponds to digit 0. After vectorizing each image into a $784 \times 1$ one-dimension signal, our imbalanced dataset contains 10 matrices $\mathbf{Y}_i \in \mathbb{R}^{784 \times 500}, i = 1, \ldots, 10$.

We first use Algorithm 4 to learn an orthogonal dictionary for each client, using their own imbalanced dataset. For client $i$, given the output of Algorithm 4 after $T$ iterations $\mathbf{D}_i^{(T)}$, we reconstruct a new signal $\mathbf{y}$ using the top $k$ atoms according to the following steps: first, we solve a *sparse coding* problem to find the sparse code $\mathbf{x}$ such that $\mathbf{y} \approx \mathbf{D}_i^{(T)} \mathbf{x}$. This can be achieved by Step 2 in Algorithm 4. Second, we find the top $k$ entries in $\mathbf{x}$ that have the largest magnitude: $\mathbf{x}_{(\alpha_1,1)}$, $\mathbf{x}_{(\alpha_2,1)}, \cdots, \mathbf{x}_{(\alpha_k,1)}$. Finally, we calculate the reconstructed signal $\tilde{\mathbf{y}}$ as

$$\tilde{\mathbf{y}} = \sum_{j=1}^k \mathbf{x}_{(\alpha_h,1)} \left( \mathbf{D}_i^{(T)} \right)_{\alpha_h}.$$

The second row of Figure 3 is generated by the above procedure with $k = 5$ using the dictionary learned by Client 1. The third row of Figure 3 corresponds to the reconstructed images using the output of PerMA.

## A.3 Details of Section 5.3

Our considered dataset in section 5.3 contains 62 frames, each of which is a $480 \times 640 \times 3$ RGB image. We consider each frame as one client ($N = 62$). After dividing each frame into $40 \times 40$ patches, we obtain each data matrix $\mathbf{Y}_i \in \mathbb{R}^{576 \times 1600}$. Then we apply PerMA to $\{\mathbf{Y}_i\}_{i=1}^{62}$ with $r_i = 576$ for all $i$ and $r^g = 30$. Consider $\mathbf{D}_i^{(T)} = \begin{bmatrix} \mathbf{D}^{g,(T)} & \mathbf{D}_i^{l,(T)} \end{bmatrix}$, which is the output of PerMA for client $i$. We reconstruct each $\mathbf{Y}_i$ using the procedure described in the previous section with $k = 50$. Specifically, we separate the contribution of $\mathbf{D}^{g,(T)}$ from $\mathbf{D}_i^{l,(T)}$. Consider the reconstructed matrix $\tilde{Y}_i$ as

$$\tilde{\mathbf{Y}}_i = \begin{bmatrix} \mathbf{D}^{g,(T)} & \mathbf{D}_i^{l,(T)} \end{bmatrix} \begin{bmatrix} \mathbf{X}_i^g \\ \mathbf{X}_i^l \end{bmatrix} = \underbrace{\mathbf{D}^{g,(T)} \mathbf{X}_i^g}_{\tilde{\mathbf{Y}}_i^g} + \underbrace{\mathbf{D}_i^{l,(T)} \mathbf{X}_i^l}_{\tilde{\mathbf{Y}}_i^l}$$

The second column and the third column of Figure 4 correspond to reconstructed results of $\tilde{\mathbf{Y}}_i^g$ and $\tilde{\mathbf{Y}}_i^l$ respectively. We can see clear separation of the background (which is shared among all frames) from the moving objects (which is unique to each frame).

One notable difference between this experiment and the previous one is in the choice of the DL algorithm $\mathcal{A}_i$. To provide more flexibility, we relax the orthogonality condition for the dictionary. Therefore, we use the alternating minimization algorithm introduced in Arora et al. (2015) for each client (see Algorithm 5). The main difference between this algorithm and Algorithm 4 is that the polar decomposition step in Algorithm 4 is replaced by a single iteration of the gradient descent applied to the loss function $\mathcal{L}(\mathbf{D}, \mathbf{X}) = \|\mathbf{D}\mathbf{X} - \mathbf{Y}\|_F^2$.

---

**Algorithm 5** Alternating minimization for general dictionary learning (Arora et al. (2015))

---

1: **Input:** $\mathbf{Y}_i, \mathbf{D}_i^{(t)}$
2: Set $\mathbf{X}_i^{(t)} = \mathrm{HT}_\zeta \left( \mathbf{D}_i^{(t)\top} \mathbf{Y}_i \right)$
3: Set $\mathbf{D}_i^{(t+1)} = \mathbf{D}_i^{(t)} - 2\eta \left( \mathbf{D}_i^{(t)} \mathbf{X}_i^{(t)} - \mathbf{Y}_i \right) \mathbf{X}_i^{(t)\top}$
4: **return** $\mathbf{D}_i^{(t+1)}$

---

Even with the computational saving brought up by Algorithm 5, the runtime significantly slows down for PerMA due to large $N$, $d$, and $p$. Here we report a practical trick to speed up PerMA, which is a local refinement procedure (Algorithm 6) added immediately before `local_update` (Step 10 of Algorithm 1). At a high level, `local_dictionary_refinement` first finds the local residual data matrix $\mathbf{Y}_i^l$ by removing the contribution of the global dictionary. Then it iteratively refines the local dictionary with respect to $\mathbf{Y}_i^l$. We observed that `local_dictionary_refinement` significantly improves the local reconstruction quality. We leave its theoretical analysis as a possible direction for future work.

---

**Algorithm 6** `local_dictionary_refinement`

---

1: **Input:** $\mathbf{D}_i^{(t)} = \begin{bmatrix} \mathbf{D}^{g,(t)} & \mathbf{D}_i^{l,(t)} \end{bmatrix}, \mathbf{Y}_i$
2: Find $\begin{bmatrix} \mathbf{X}_i^g \\ \mathbf{X}_i^l \end{bmatrix}$ such that $\mathbf{Y}_i \approx \begin{bmatrix} \mathbf{D}^{g,(t)} & \mathbf{D}_i^{l,(t)} \end{bmatrix} \begin{bmatrix} \mathbf{X}_i^g \\ \mathbf{X}_i^l \end{bmatrix}$
   $\qquad\qquad\qquad\qquad\qquad\qquad\qquad$ // Solving a sparse coding problem
3: Set $\mathbf{Y}_i^l = \mathbf{Y}_i - \mathbf{D}^{g,(t)} \mathbf{X}_i^g$
4: Set $\mathbf{D}_i^{\mathrm{refine},(0)} = \mathbf{D}_i^{l,(t)}$.
5: **for** $\tau = 0, 1, ..., T^{\mathrm{refine}} - 1$ **do**
6: $\quad$ Set $\mathbf{D}_i^{\mathrm{refine},(\tau+1)} = \mathcal{A}_i \left( \mathbf{Y}_i^l, \mathbf{D}_i^{\mathrm{refine},(\tau)} \right)$ $\qquad$ // Improving local dictionary
7: **end for**
8: **return** $\mathbf{D}_i^{\mathrm{refine},(T^{\mathrm{refine}})}$ as refined $\mathbf{D}_i^{l,(t)}$

---

# B Additional Experiments

In this section, we present the results of some additional experiments to better showcase the efficiency of PerMA compared with the existing methods and its potential to adapt to a parameter-free algorithm.

## B.1 Comparison with Existing Methods

To the best of our knowledge, the problem of personalized dictionary learning (PerDL) has not been previously explored and formally defined. While existing methods in federated learning bear some resemblance to PerDL, they lack provable guarantees in recovering the global and local dictionaries. To clarify these distinctions, we present a detailed comparison between our work and the most closely related papers by Huang et al. (2022) and Gkillas et al. (2022). We compare our method with these methods under the same setting as in Section 5.1. The results can be seen in Figure 5, which shows that PerMA consistently outperforms methods proposed by Gkillas et al. (2022) and Huang et al. (2022).

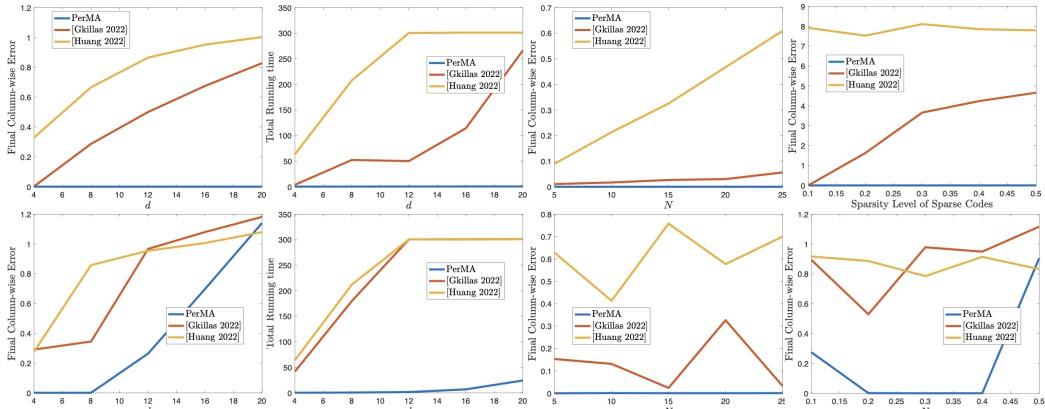

Figure 5: Comparisons of different methods on synthetic datasets. In the first row, clients are provided with heterogeneous datasets with similar sizes; in the second row, we consider the special cases in which one of the clients has an insufficient sample size and evaluate the performance of the dictionary learned by that specific client. The first column corresponds to final errors with varying $d$; the second column corresponds to total running times with varying $d$; the third column corresponds to final errors with varying $N$; and the forth column corresponds to final errors with varying sparsity level. All the results are averaged over 3 independent trials.

Next, in the context of Section 5.2, we compare the quality of the reconstructed images using dictionaries learned from different methods under three metrics: MSE, PSNR and SSIM. A smaller MSE, a larger PSNR, and an SSIM closer to 1 indicate better image reconstruction quality. In Table 1, $k$ denotes the number of atoms used to reconstruct the image. As can be seen in the table, PerMA achieves the best result in all sections except for the training time.

Finally, in this paper we use PerMA on the surveillance video datasets, with the goal of separating common elements shared by all clients (the background) and unique elements (different cars). Such a task cannot be accomplished by Gkillas et al. (2022) and Huang et al. (2022) due to their lack of personalization. On the other hand, a recently proposed method based on personalized PCA (PerPCA) has been shown to be effective in separating common and unique elements (Shi and Kontar, 2022). As a result, we run PerPCA on the same dataset to compare its performance with our method. According to Figure 6, PerDL outperforms PerPCA by achieving better separation and higher resolution.

## B.2 Auto-tuning of $r_g$

One can easily see from Section 3 that $r_g$ is an important hyper-parameter of our algorithm. A larger $r_g$ means more global atoms are sent between central servers and clients, which leads to a stronger collaboration between them. In the synthetic experiment, we assume to know the value of $r_g$, while in

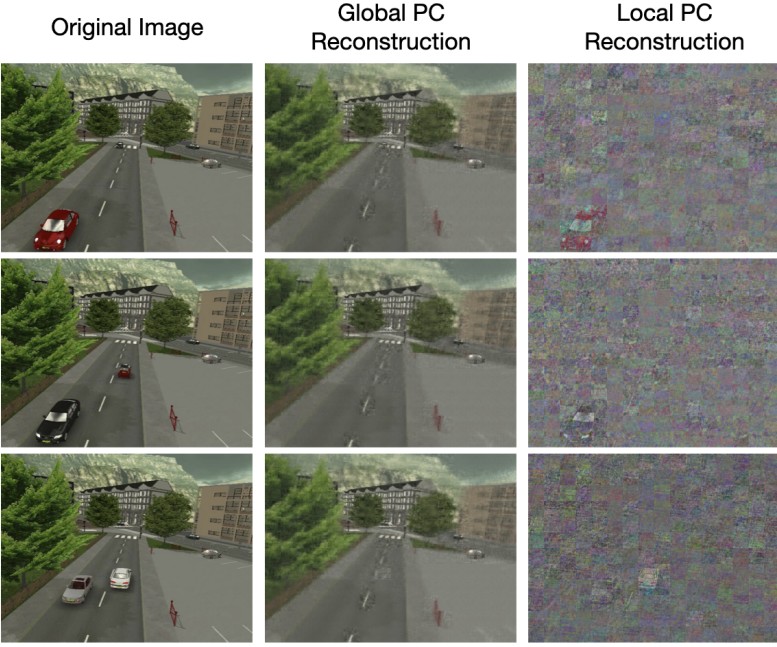

Figure 6: Objects and background separation by PerPCA.

real-life applications, one needs to fine-tune this parameter. An interesting observation we made is that the proposed PerMA algorithm can be augmented by a simple detection mechanism for identifying the correct choice of $r_g$. Specifically, during the Global Matching step, where we iteratively remove shortest paths, we can monitor the length of the obtained shortest path. By terminating the removal of paths (i.e., adding global atoms) when the length of the path experiences a significant increase beyond a predefined threshold, we can effectively identify the appropriate value of $r_g$ without requiring prior knowledge. This detection mechanism alleviates the burden of fine-tuning $r_g$ and allows for a more practical and robust implementation of the algorithm.

To validate the efficacy of this approach, we conducted a series of experiments, the results of which are presented in Figure 7. We use different $r_g = 4, 6, 8$ with $r = 10$ and monitor the lengths of paths. As evident from the outcomes, a clear and drastic increase in the length of the $r_g + 1$-th shortest path is observed, signifying the correct value of $r_g$.

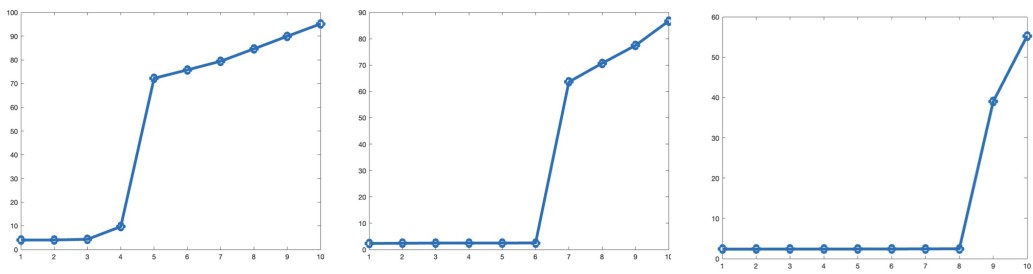

Figure 7: Increase of path length for different $r_g$. Here the $x$-axis denotes the number of iterations for Algorithm 2 and the $x$-axis denotes the distance of the $\mathcal{P}$ for each iteration.

| Methods | MSE | | PSNR | | SSIM | | Running Time |
|---|---|---|---|---|---|---|---|
| | $k = 10$ | $k = 20$ | $k = 10$ | $k = 20$ | $k = 10$ | $k = 20$ | (seconds) |
| PerMA | 0.0319 | 0.0207 | 15.3795 | 17.3771 | 0.6286 | 0.7074 | 40.8 |
| Local DL | 0.0448 | 0.0330 | 14.1936 | 15.7203 | 0.5538 | 0.6162 | 4.9 |
| Gkillas et al. | 0.1069 | 0.1084 | 9.8930 | 9.8360 | 0.2736 | 0.2659 | 3600+ |
| Huang et al. | 0.1062 | 0.1062 | 9.9203 | 9.9212 | 0.2737 | 0.2628 | 3600+ |
| PerPCA | 0.0837 | 0.0734 | 11.0118 | 11.6426 | 0.3520 | 0.3959 | 59.3 |

Table 1: Comparisons of different methods on MNIST datasets. A smaller MSE, a larger PSNR, and an SSIM closer to 1 indicate better image reconstruction quality. Here $k$ denotes the number of atoms used to reconstruct the image.

## C   Further Discussion on Linearly Convergent Algorithms

In this section, we discuss a linearly convergent DL algorithm that satisfies the conditions of our Theorem 2. In particular, the next theorem is adapted from (Arora et al., 2015, Theorem 12) and shows that a modified variant of Algorithm 5 introduced in (Arora et al., 2015, Algorithm 5) is indeed linearly-convergent.

**Theorem 3** (Linear convergence of Algorithm 5 in Arora et al. (2015)). *Suppose that the data matrix satisfies* $\mathbf{Y} = \mathbf{D}^*\mathbf{X}^*$, *where* $\mathbf{D}^*$ *is an* $\mu$-*incoherent dictionary and the sparse code* $\mathbf{X}^*$ *satisfies the generative model introduced in Section 1.2 and Section 4.1 of Arora et al. (2015). For any initial dictionary* $\|\mathbf{D}^{(0)}\|_2 \leq 1$, *Algorithm 5 in Arora et al. (2015) is* $(\delta, \rho, \psi)$-*linearly convergent with* $\delta = O(1/\log d)$, $\rho \in (1/2, 1)$, *and* $\psi = O(d^{-\omega(1)})$.

Algorithm 5 in Arora et al. (2015) is a refinement of Algorithm 5, where the error is further reduced by projecting out the components along the column currently being updated. For brevity, we do not discuss the exact implementation of the algorithm; an interested reader may refer to Arora et al. (2015) for more details. Indeed, we have observed in our experiments that the additional projection step does not provide a significant benefit over Algorithm 5.

## D   Proof of Theorems

### D.1   Proof of Theorem 1

To begin with, we establish a triangular inequality for $d_{1,2}(\cdot, \cdot)$, which will be important in our subsequent arguments:

**Lemma 1** (Triangular inequality for $d_{1,2}(\cdot, \cdot)$). *For any dictionary* $\mathbf{D}_1, \mathbf{D}_2, \mathbf{D}_3 \in \mathbb{R}^{d \times r}$, *we have*

$$d_{1,2}(\mathbf{D}_1, \mathbf{D}_2) \leq d_{1,2}(\mathbf{D}_1, \mathbf{D}_3) + d_{1,2}(\mathbf{D}_3, \mathbf{D}_2) \tag{13}$$

*Proof.* Suppose $\mathbf{\Pi}_{1,3}$ and $\mathbf{\Pi}_{3,2}$ satisfy $d_{1,2}(\mathbf{D}_1, \mathbf{D}_3) = \|\mathbf{D}_1\mathbf{\Pi}_{1,3} - \mathbf{D}_3\|_{1,2}$ and $d_{1,2}(\mathbf{D}_3, \mathbf{D}_2) = \|\mathbf{D}_3 - \mathbf{D}_2\mathbf{\Pi}_{3,2}\|_{1,2}$. Then we have

$$\begin{aligned} d_{1,2}(\mathbf{D}_1, \mathbf{D}_3) + d_{1,2}(\mathbf{D}_3, \mathbf{D}_2) &= \|\mathbf{D}_1\mathbf{\Pi}_{1,3} - \mathbf{D}_3\|_{1,2} + \|\mathbf{D}_3 - \mathbf{D}_2\mathbf{\Pi}_{3,2}\|_{1,2} \\ &\geq \|\mathbf{D}_1\mathbf{\Pi}_{1,3} - \mathbf{D}_2\mathbf{\Pi}_{3,2}\|_{1,2} \\ &\geq d_{1,2}(\mathbf{D}_1, \mathbf{D}_2). \end{aligned} \tag{14}$$

$\square$

Given how the directed graph $\mathcal{G}$ is constructed and modified, any directed path from $s$ to $t$ will be of the form $\mathcal{P} = s \to (\mathbf{D}_1^{(0)})_{\alpha(1)} \to (\mathbf{D}_2^{(0)})_{\alpha(2)} \to \cdots \to (\mathbf{D}_N^{(0)})_{\alpha(N)} \to t$. Specifically, each layer (or client) contributes exactly one node (or atom), and the path is determined by $\alpha(\cdot) : [N] \to [r]$. Recall that $\mathbf{D}_i^* = \begin{bmatrix} \mathbf{D}^{g*} & \mathbf{D}_i^{l*} \end{bmatrix}$ for every $1 \leq i \leq N$. Assume, without loss of generality, that for every client $1 \leq i \leq N$,

$$\mathbf{I}_{r_i \times r_i} = \arg\min_{\mathbf{\Pi} \in \mathcal{P}(r_i)} \left\| \mathbf{D}_i^* \mathbf{\Pi} - \mathbf{D}_i^{(0)} \right\|_{1,2}. \tag{15}$$

In other words, the first $r^g$ atoms in the initial dictionaries $\{D_i^{(0)}\}_{i=1}^N$ are aligned with the global dictionary. Now consider the special path $\mathcal{P}_j^*$ for $1 \leq j \leq r^g$ defined as

$$\mathcal{P}_j^* = s \to (\mathbf{D}_1^{(0)})_j \to (\mathbf{D}_2^{(0)})_j \to \cdots \to (\mathbf{D}_N^{(0)})_j \to t. \tag{16}$$

To prove that Algorithm 2 correctly selects and aligns global atoms from clients, it suffices to show that $\{\mathcal{P}_j^*\}_{j=1}^{r^g}$ are the top-$r^g$ shortest paths from $s$ to $t$ in $\mathcal{G}$. The length of the path $\mathcal{P}_j^*$ can be bounded as

$$
\begin{aligned}
\mathcal{L}\left(\mathcal{P}_j^*\right) &= \sum_{i=1}^{N-1} d_2\left((\mathbf{D}_i^{(0)})_j, (\mathbf{D}_{i+1}^{(0)})_j\right) \\
&= \sum_{i=1}^{N-1} \min\left\{\|(\mathbf{D}_i^{(0)})_j - (\mathbf{D}_{i+1}^{(0)})_j\|_2, \|(\mathbf{D}_i^{(0)})_j + (\mathbf{D}_{i+1}^{(0)})_j\|_2\right\} \\
&\leq \sum_{i=1}^{N-1} \|(\mathbf{D}_i^{(0)})_j - (\mathbf{D}_{i+1}^{(0)})_j\|_2 \\
&\leq \sum_{i=1}^{N-1} \|(\mathbf{D}_i^{(0)})_j - (\mathbf{D}^{g*})_j\|_2 + \|(\mathbf{D}_{i+1}^{(0)})_j - (\mathbf{D}^{g*})_j\|_2 \\
&\leq \sum_{i=1}^{N-1} (\epsilon_i + \epsilon_{i+1}) \\
&\leq 2\sum_{i=1}^N \epsilon_i.
\end{aligned}
\tag{17}
$$

We move on to prove that all the other paths from $s$ to $t$ will have a distance longer than $2\sum_{i=1}^N \epsilon_i$. Consider a general directed path $\mathcal{P} = s \to (\mathbf{D}_1^{(0)})_{\alpha(1)} \to (\mathbf{D}_2^{(0)})_{\alpha(2)} \to \cdots \to (\mathbf{D}_N^{(0)})_{\alpha(N)} \to t$ that is not in $\{\mathcal{P}_j^*\}_{j=1}^{r^g}$. Based on whether or not $\mathcal{P}$ contains atoms that align with the true global ground atoms, there are two situations:

**Case 1:** Suppose there exists $1 \leq i \leq N$ such that $\alpha(i) \leq r^g$. Given Model 1 and the assumed equality (15), we know that for layer $i$, $\mathcal{P}$ contains a global atom. Since $\mathcal{P}$ is not in $\{\mathcal{P}_j^*\}_{j=1}^{r^g}$, there must exist $k \neq i$ such that $\alpha(k) \neq \alpha(i)$. As a result, we have

$$
\begin{aligned}
\mathcal{L}(\mathcal{P}) &\overset{(a)}{\geq} d_{1,2}\left((\mathbf{D}_i^{(0)})_{\alpha(i)}, (\mathbf{D}_k^{(0)})_{\alpha(k)}\right) \\
&\overset{(b)}{\geq} \min\left\{\|(\mathbf{D}_i^*)_{\alpha(i)} - (\mathbf{D}_k^*)_{\alpha(k)}\|_2, (\mathbf{D}_i^*)_{\alpha(i)} + (\mathbf{D}_k^*)_{\alpha(k)}\|_2\right\} \\
&\quad - \|(\mathbf{D}_i^*)_{\alpha(i)} - (\mathbf{D}_i^{(0)})_{\alpha(i)}\|_2 - \|(\mathbf{D}_k^*)_{\alpha(k)} - (\mathbf{D}_k^{(0)})_{\alpha(k)}\|_2 \\
&\overset{(c)}{\geq} \sqrt{2 - 2\left|\langle(\mathbf{D}_k^*)_{\alpha(i)}, (\mathbf{D}_k^*)_{\alpha(k)}\rangle\right|} - 2\max_{1 \leq i \leq N} \epsilon_i \\
&\overset{(d)}{\geq} \sqrt{2 - 2\frac{\mu}{\sqrt{d}}} - 2\max_{1 \leq i \leq N} \epsilon_i \\
&\overset{(e)}{\geq} 2\sum_{i=1}^N \epsilon_i^g
\end{aligned}
\tag{18}
$$

Here $(a)$ and $(b)$ are due to Lemma 1, $(c)$ is due to assumed equality (15), $(d)$ is due to the $\mu$-incoherency of $\mathbf{D}_k^*$, and finally $(e)$ is given by the assumption of Theorem 1.

**Case 2:** Suppose $\alpha(i) > r^g$ for all $1 \leq i \leq N$, which means that the path $\mathcal{P}$ only uses approximations to local atoms. Consider the ground truth of these approximations, $(\mathbf{D}_1^*)_{\alpha(1)}, (\mathbf{D}_2^*)_{\alpha(2)}, ..., (\mathbf{D}_N^*)_{\alpha(N)}$. There must exist $1 \leq i, j \leq N$ such that $d_{1,2}\left((\mathbf{D}_i^*)_{\alpha(i)}, (\mathbf{D}_j^*)_{\alpha(j)}\right) \geq \beta$. Otherwise, it is easy to see that $\{\mathbf{D}_i^{l*}\}_{i=1}^N$ would not be $\beta$-identifiable

because any $(\mathbf{D}_i^*)_{\alpha(i)}$ will satisfy (6). As a result, we have the following:

$$
\begin{aligned}
\mathcal{L}(\mathcal{P}) &\geq d_{1,2}\left((\mathbf{D}_i^{(0)})_{\alpha(i)}, (\mathbf{D}_j^{(0)})_{\alpha(j)}\right) \\
&\geq d_{1,2}\left((\mathbf{D}_i^*)_{\alpha(i)}, (\mathbf{D}_j^*)_{\alpha(j)}\right) - \|(\mathbf{D}_i^*)_{\alpha(i)} - (\mathbf{D}_i^{(0)})_{\alpha(i)}\|_2 - \|(\mathbf{D}_j^*)_{\alpha(j)} - (\mathbf{D}_j^{(0)})_{\alpha(j)}\|_2 \\
&\geq \beta - 2\max_i \epsilon_i \\
&\geq 2\sum_{i=1}^N \epsilon_i
\end{aligned}
\tag{19}
$$

So we have shown that $\{\mathcal{P}_j^*\}_{j=1}^{r^g}$ are the top-$r^g$ shortest paths from $s$ to $t$ in $\mathcal{G}$. Moreover, it is easy to show that $\mathrm{sign}\left(\left\langle (\mathbf{D}_1^{(0)})_j, (\mathbf{D}_i^{(0)})_j \right\rangle\right) = 1$ for small enough $\{\epsilon_i\}_{i=1}^N$. Therefore, the proposed algorithm correctly recovers the global dictionaries (with the correct identity permutation). Finally, we have $\mathbf{D}^{g,(0)} = \frac{1}{N}\sum_{i=1}^N (\mathbf{D}_i^{(0)})_{1:r^g}$, which leads to:

$$
\begin{aligned}
d_{1,2}\left(\mathbf{D}^{g,(0)}, \mathbf{D}^{g*}\right) &\leq \max_{1\leq j\leq r^g} \left\| \frac{1}{N}\sum_{i=1}^N (\mathbf{D}_i^{(0)})_j - (\mathbf{D}^{g*})_j \right\|_2 \\
&\leq \max_{1\leq j\leq r^g} \frac{1}{N}\sum_{i=1}^N \left\| (\mathbf{D}_i^{(0)})_j - (\mathbf{D}^{g*})_j \right\|_2 \\
&\leq \max_{1\leq j\leq r^g} \frac{1}{N}\sum_{i=1}^N \epsilon_i \\
&= \frac{1}{N}\sum_{i=1}^N \epsilon_i.
\end{aligned}
\tag{20}
$$

This completes the proof of Theorem 1. $\qquad\square$

### D.2 Proof of Theorem 2

Throughout this section, we define:

$$
\bar{\rho} := \frac{1}{N}\sum_{i=1}^N \rho_i, \qquad \bar{\psi} := \frac{1}{N}\sum_{i=1}^N \psi_i.
\tag{21}
$$

We will prove the convergence of the global dictionary in Theorem 2 by proving the following induction: at each $t \geq 1$, we have

$$
d_{1,2}\left(\mathbf{D}^{g,(t+1)}, \mathbf{D}^{g*}\right) \leq \bar{\rho}\, d_{1,2}\left(\mathbf{D}^{g,(t)}, \mathbf{D}^{g*}\right) + \bar{\psi}.
\tag{22}
$$

At the beginning of communication round $t$, each client $i$ performs `local_update` to get $\mathbf{D}_i^{(t+1)}$ given $\begin{bmatrix} \mathbf{D}^{g,(t)} & \mathbf{D}_i^{l,(t)} \end{bmatrix}$. Without loss of generality, we assume

$$
\mathbf{I}_{r_i \times r_i} = \arg\min_{\mathbf{\Pi}\in\mathcal{P}(r_i)} \left\| \mathbf{D}_i^*\mathbf{\Pi} - \begin{bmatrix} \mathbf{D}^{g,(t)} & \mathbf{D}_i^{l,(t)} \end{bmatrix} \right\|_{1,2},
\tag{23}
$$

$$
\mathbf{I}_{r_i \times r_i} = \arg\min_{\mathbf{\Pi}\in\mathcal{P}(r_i)} \left\| \mathbf{D}_i^*\mathbf{\Pi} - \mathbf{D}_i^{(t+1)} \right\|_{1,2}.
\tag{24}
$$

Assumed equalities (23) and (24) imply that the permutation matrix that aligns the input and the output of $\mathcal{A}_i$ is also $\mathbf{I}_{r_i \times r_i}$. Specifically, the linear convergence property of $\mathcal{A}_i$ and Theorem 1 thus suggest:

$$
\left\| \left(\mathbf{D}_i^{(t+1)}\right)_j - (\mathbf{D}_i^*)_j \right\|_2 \leq \rho_i \left\| \left(\mathbf{D}^{g,(t)}\right)_j - (\mathbf{D}_i^*)_j \right\|_2 + \psi_i \quad \forall 1 \leq j \leq r^g, 1 \leq i \leq N.
\tag{25}
$$

However, our algorithm is unaware of this trivial alignment. We will next show the remaining steps in `local_update` correctly recovers the identity permutation. The proof is very similar to the proof of Theorem 1 since we are essentially running Algorithm 2 on a two-layer $\mathcal{G}$. For every $1 \leq i \leq N$, $1 \leq j \leq r^g$, we have

$$d_{1,2}\left(\left(\mathbf{D}_i^{(t+1)}\right)_j, \left(\mathbf{D}^{g,(t)}\right)_j\right) \leq d_{1,2}\left(\left(\mathbf{D}_i^{(t+1)}\right)_j, (\mathbf{D}_i^*)_j\right) + d_{1,2}\left((\mathbf{D}_i^*)_j, \left(\mathbf{D}^{g,(t)}\right)_j\right) \quad (26)$$

$$\leq 2\delta_i.$$

Meanwhile for $k \neq j$,

$$d_{1,2}\left(\left(\mathbf{D}_i^{(t+1)}\right)_k, \left(\mathbf{D}^{g,(t)}\right)_j\right)$$

$$\geq d_{1,2}\left((\mathbf{D}_i^*)_k, (\mathbf{D}_i^*)_j\right) - d_{1,2}\left(\left(\mathbf{D}_i^{(t+1)}\right)_k, (\mathbf{D}_i^*)_k\right) - d_{1,2}\left((\mathbf{D}_i^*)_j, \left(\mathbf{D}^{g,(t)}\right)_j\right) \quad (27)$$

$$\geq \sqrt{2 - \frac{2\mu}{\sqrt{d}}} - 2\delta_i.$$

$$\geq 2\delta_i.$$

As a result, we successfully recover the identity permutation, which implies

$$\left\|\left(\mathbf{D}_i^{g,(t+1)}\right)_j - (\mathbf{D}_i^{g*})_j\right\|_2 \leq \rho_i \left\|\left(\mathbf{D}^{g,(t)}\right)_j - (\mathbf{D}_i^{g*})_j\right\|_2 + \psi_i \quad \forall 1 \leq j \leq r^g, 1 \leq i \leq N. \quad (28)$$

Finally, the aggregation step (Step 13 in Algorithm 1) gives:

$$d_{1,2}\left(\mathbf{D}^{g,(t+1)}, \mathbf{D}^{g*}\right) \leq \left\|\frac{1}{N}\sum_{i=1}^{N}\mathbf{D}_i^{g,(t+1)} - \mathbf{D}^{g*}\right\|_{1,2}$$

$$= \max_{1 \leq j \leq r^g}\left\|\left(\frac{1}{N}\sum_{i=1}^{N}\mathbf{D}_i^{g,(t+1)}\right)_j - (\mathbf{D}^{g*})_j\right\|$$

$$\leq \max_{1 \leq j \leq r^g}\frac{1}{N}\sum_{i=1}^{N}\left\|\left(\mathbf{D}_i^{g,(t+1)}\right)_j - (\mathbf{D}_i^{g*})_j\right\|_2 \quad (29)$$

$$\leq \max_{1 \leq j \leq r^g}\frac{1}{N}\sum_{i=1}^{N}\left(\rho_i\left\|\left(\mathbf{D}^{g,(t)}\right)_j - (\mathbf{D}_i^{g*})_j\right\|_2 + \psi_i\right)$$

$$\leq \frac{1}{N}\sum_{i=1}^{N}\left(\rho_i d_{1,2}\left(\mathbf{D}^{g,(t)}, \mathbf{D}^{g*}\right) + \psi_i\right)$$

$$= \bar{\rho}d_{1,2}\left(\mathbf{D}^{g,(t)}, \mathbf{D}^{g*}\right) + \bar{\psi}.$$

As a result, we prove the induction (22) for all $0 \leq t \leq T - 1$. Inequality (12) is a by-product of the accurate separation of global and local atoms and can be proved by similar arguments. The proof is hence complete. $\qquad\square$

