# OpenReview forum: "Personalized Dictionary Learning for Heterogeneous Datasets"
_NeurIPS.cc/2023/Conference — NeurIPS 2023 poster_

### Official Review · Reviewer_ED5i · 2023-07-02

**Soundness:** 2 fair
**Presentation:** 3 good
**Contribution:** 2 fair
**Rating:** 4
**Confidence:** 5

**Summary:**

This paper tackles the problem of personalized dictionary learning (PerDL) for heterogeneous datasets that share some commonality. The authors propose a federated meta-algorithm called PerMA that can provably recover both global and local dictionaries from heterogeneous datasets. They show the applications of PerDL in different learning tasks, such as training with imbalanced datasets and video surveillance.

--post rebuttal
After reading the rebuttal, I would like to keep my score.


**Strengths:**

•	The paper proposes a federated meta-algorithm called PerMA that can provably recover both global and local dictionaries from heterogeneous datasets.
•	The paper provides theoretical guarantees on the identifiability, convergence, and robustness of PerMA, and demonstrates its applications in different learning tasks. The result on the Surveillance Video Dataset looks interesting.


**Weaknesses:**

•	The paper does not compare PerMA with other existing methods for dictionary learning or federated learning, such as personalized PCA (PerPCA), which would be helpful to evaluate its performance and advantages.
•	In Surveillance Video experiments, why there is only one original image and one local atom? The author is encouraged to give some explanation. Moreover, only one case study seems insufficient.
•	Federated Learning is a long studying problem so far, the author is needed to further highlight the difference between their proposed ``global matching and local update” with existing methods.


**Questions:**

My biggest concern is about the insufficient experiments. This paper lacks a comparison between existing dictionary learning methods. Moreover, there is no actual numerical result in experiments. I’d like to see more experiments during the rebuttal.

---

> ### Author Rebuttal · Authors · 2023-08-09
>
> The authors truly appreciate the reviewer's insightful comments.
>
> > The paper does not compare PerMA with other existing methods for dictionary learning or federated learning, such as personalized PCA (PerPCA), which would be helpful to evaluate its performance and advantages.
>
> We kindly refer the reviewer to our general response, where we validate the effectiveness and efficiency of our method compared to various state-of-the-art methods (including PerPCA). We will add these new experiments to our revised paper.
>
> >In Surveillance Video experiments, why there is only one original image and one local atom? The author is encouraged to give some explanation. Moreover, only one case study seems insufficient.
>
> There seems to be a misunderstanding in our experimental setup, and we apologize for any confusion caused. To clarify, there is indeed more than one local and global atom corresponding to each frame. During our training process, there are in total $61$ different images that share common backgrounds but have individually unique components (cars). For each frame, we learned 30 global atoms and 546 local atoms. The pictures on the second and third columns are reconstructions of the original images using only 50 atoms of the learned dictionary. We apologize for the lack of clarity in our explanation, which will be addressed in the revised paper.
>
> Regarding the reviewer's claim about the numerical section, we want to clarify that our work indeed includes three case studies, and two of these case studies specifically pertain to real-life applications in image and video processing.
>
> > Federated Learning is a long studying problem so far, the author is needed to further highlight the difference between their proposed ``global matching and local update” with existing methods.
>
> We express our gratitude to the reviewer for raising this important comment. It is crucial to emphasize that, to the best of our knowledge, the problem of personalized dictionary learning (PerDL) has not been previously explored and formally defined. While existing methods in federated learning bear some resemblance to PerDL, they are not directly applicable to address this problem with provable guarantees. To clarify these distinctions, we present a detailed comparison between our work and the most closely related papers by *Huang et al. (2022)* and *Gkillas et al.(2022)*.
>
> Firstly, we rigorously define the problem of personalized dictionary learning in our paper, and our proposed method PerMA is explicitly tailored to extract commonality while preserving heterogeneity across clients. In *Huang et al. (2022)*, no personalization is considered, and in *Gkillas et al.(2022)*, their solution to personalization involves merely broadcasting frequently used atoms without a comprehensive approach to address the underlying challenges.
>
> Secondly, a critical aspect in solving PerDL is the alignment of atoms from different clients, and we address this challenge through our novel graph-based method. In contrast, *Huang et al. (2022)* proposes a brute-force approach for atom alignment, while *Gkillas et al.(2022)* does not consider this important aspect at all.
>
> Thirdly, existing works on federated learning, including the two papers mentioned above, have a broader scope and lack specific guarantees on PerDL. In contrast, our meta-algorithm is explicitly designed to tackle PerDL and is equipped with provable guarantees, ensuring its efficacy and effectiveness in personalized dictionary learning scenarios.
>
> Finally, we kindly refer the reviewer to our general response, where we provide a detailed comparison between the performance of our method and these methods.
>
> ### Reference
>
> Gkillas, A., Ampeliotis, D., and Berberidis, K. (2022). Federated dictionary learning from non-iid data. In 2022 IEEE 14th Image, Video, and Multidimensional Signal Processing Workshop(IVMSP), pages 1–5. IEEE.
>
> Huang, K., Liu, X., Li, F., Yang, C., Kaynak, O., and Huang, T. (2022). A federated dictionary learning method for process monitoring with industrial applications. IEEE Transactions on Artificial Intelligence.

---

### Official Review · Reviewer_zsPp · 2023-07-04

**Soundness:** 3 good
**Presentation:** 3 good
**Contribution:** 3 good
**Rating:** 6
**Confidence:** 3

**Summary:**

This work tackles the problem of heterogeneity in federated learning through dictionary learning. The authors name this problem _Personalized Dictionary Learning_ (PerDL), which seeks to learn (linear) representations for the heterogeneous datasets from clients, which are supposed to share common characteristics. The insight of their approach is that PerDL will disentangle global/general and local/unique features from clients via DL. The authors provide a thorough analysis of convergence of their method, as well as a federated strategy for learning the dictionaries (PerMA).

__Post-Rebuttal Acknowledgment__

I have read the authors rebuttal, and other reviewers comments. In their rebuttal, the authors fully addressed my concerns, and provided important results and discussion, including further discussion on the ethical impacts of their work. Overall, I think the authors rebuttal greatly improves the quality of their submission. As a result, I increased my score to 6. Weak Accept.

**Strengths:**

__Originality.__ The authors present an original and theoretically grounded work for dictionary learning in a federated setting.

__Quality and Clarity.__ The paper is well written and clear. Assumptions are clearly stated and the concepts behind theorems are clearly defined. Analyzing the abstract alone, the federated aspect of this work is not clear (see W1 and S4 below).

__Significance.__ While I consider the authors contribution novel, the fact that in the paper only qualitative results are shown, and that the authors do not draw comparisons with other SOTA methods hinders the significance of this work.

**Weaknesses:**

Below I list a series of weaknesses of the current paper. Please see my suggestions in the next section on how to improve these points.

__Concerning Clarity__

__W1.__ Even though it is clear from the introduction, the abstract does not mention federated learning at all.

__Concerning the authors experiments__

__W2.__ Sections 5.2 only contain qualitative results (i.e., Figures 3 and 4). This hinders the assessment of their method, especially in comparison to other strategies.

__W3.__ The authors only provide comparisons with non-collaborative/non-federated approaches. A comparison with the methods of [Gkillas et al., 2022] and/or [Huang et al., 2022] would improve the impact of the authors results.

__Concerning Ethical Considerations__

__W4.__ Even though the authors method is not especifically tailored for video surveillance, I think that authors should provide a broader discussion on the ethical impacts of their experiment.

__Post Rebuttal__

In their rebuttal, the authors correctly addressed all of the above weaknesses. As a result I raised my score from 3. Reject towards 6. Weak Accept.

**Questions:**

## Suggestions

Here, authors may find a list of suggestions associated with the raised weaknesses.

__S1.__ Authors should include the federated learning motivation in their abstract.

__S2.__ Include quantitative results in Sections 5.2 and 5.3. Concerning image reconstruction, a table comparing PSNR, MSE or SSIM metrics of the different tested strategies would be intersting. The same metrics would apply to the data of individual clients. Concerning surveillance video, the authors could consider the F-score as done in [Cuevas et al., 2016, Table 7].

__S3.__ Compare their method against SOTA methods (e.g., [Gkillas et al., 2022] and/or [Huang et al., 2022])

__S4.__ Include a discussion about the possible impacts of the authors work in video surveillance.

**Limitations:**

The authors did not explicitly discuss the limitations of their work in the main paper. Some discussion is given in the appendix A.3, where the authors comment on the performance of their method when $N$, $d$ and $p$ are large. Furthermore, in section 3.1 the authors analyze the complexity of their method. I think the authors could contextualize this discussion with the overall challenges of large scale tasks (such as video surveillance).

---

> ### Author Rebuttal · Authors · 2023-08-09
>
> We really appreciate the reviewer's helpful comments and detailed suggestions.
>
> > W1/S1
>
> We thank the reviewer for this comment and apologize for not mentioning federated learning in the abstract. Rest assured, we will address this issue by adding federated learning motivation to our abstract.
>
> >W2/W3/S2/S3
>
> We acknowledge and apologize for the insufficiency of comparisons and quantification in our numerical sections. We kindly refer the reviewer to our general response, where we validate the effectiveness and efficiency of our method compared to various SOTA methods (including those mentioned by the reviewer) and with quantifiable metrics. We will add these new experiments to our revised paper.
>
> >W4/S4
>
> We thank the reviewer for raising their concern regarding the ethical impact of our work. This concern has been further investigated by two additional ethics reviewers, and both indicated that neither our method nor our simulations raise any significant ethical issues.
>
> Nonetheless, we plan to add the following discussion on the broader and social impact of our work:
>
> "Our novel approach for personalized dictionary learning presents a versatile solution with immediate applications across various domains, such as video surveillance and object detection. While these applications offer valuable benefits, they also bring to the forefront ethical and societal concerns, particularly concerning privacy, bias, and potential misuse.
>
> In the context of video surveillance, deploying object detection algorithms may inadvertently capture private information, leading to concerns about violating individuals' right to privacy. However, it is important to emphasize that our proposed algorithm is specifically designed to focus on separating unique and common features within the data, without delving into the realm of personal information. Thus, it adheres to ethical principles by ensuring that private data is not processed or used without explicit consent.
>
> Bias is another critical aspect that necessitates careful consideration in the deployment of object detection algorithms. Biases can manifest in various forms, such as underrepresentation or misclassification of certain groups, leading to discriminatory outcomes. Our approach acknowledges the importance of mitigating biases by solely focusing on the distinction between common and unique features, rather than introducing any inherent bias into the learning process.
>
> Furthermore, the potential misuse of object detection algorithms in unauthorized surveillance or invasive tracking scenarios raises valid concerns. As responsible researchers, we are cognizant of such risks and stress that our proposed algorithm is meant to be deployed in a controlled and legitimate manner, adhering to appropriate regulations and ethical guidelines."
>
>
> > The authors did not explicitly discuss the limitations of their work in the main paper.
>
> We thank the reviewer for their comment. To address it, we will add the following paragraph to the paper:
>
> "Even though our meta-algorithm PerMA enjoys strong theoretical guarantees and practical performance, there are still several avenues for improvement. For instance, the theoretical success of PerMA, especially the Global Matching step, relies on an individual initial error of $O(1/N)$. In other words, the initial error should decrease as the number of clients grows. As a future work, we plan to relax such dependency via a more delicate analysis. We also note that imposing an upper bound on the initial error is not unique to our setting, as virtually all existing algorithms for classical (non-personalized) dictionary learning require certain conditions on the initial error.  On the other hand, once the assumption on the initial error is satisfied, our meta-algorithm achieves a final error with the same dependency on $d$ (the dimensionality of the data) and $n$ (the number of samples) as the state-of-the-art algorithms for classical dictionary learning (*Agarwal et al. (2016)*, *Arora et al. (2015)*). Remarkably, this implies that personalization is achieved without incurring any additional cost on $d$ or $n$, making PerMA highly efficient and competitive in its performance."
>
> ### Reference
>
> Agarwal, A., Anandkumar, A., Jain, P., and Netrapalli, P. (2016). Learning sparsely used overcomplete dictionaries via alternating minimization. SIAM Journal on Optimization, 26(4):2775–2799.
>
> Arora, S., Ge, R., Ma, T., and Moitra, A. (2015). Simple, efficient, and neural algorithms for sparse coding. In Conference on learning theory, pages 113–149. PMLR.

---

> > ### Comment · Reviewer_zsPp · 2023-08-13
> >
> > I thank the authors for their detailed rebuttal, their additional experiments and the discussion on the ethical impact of their work. As a result of these elements I am raising my score from 3. Rejection towards 6. Weak Accept.
> >
> > Overall, the authors correctly addressed the issues raised in my review. Especially, authors included new quantitative results, and compared their method to existing state-of-the-art. This significantly improves the significance of their experimental section. Furthermore, the authors included an important discussion on the ethical impacts of their experiments.

---

> > > ### Author Response · Authors · 2023-08-14
> > >
> > > We thank the reviewer for carefully reading our rebuttal and raising the score. We would also like to thank the reviewer again for the helpful suggestions on improving our paper, especially the numerical section. We will include all the points raised during the rebuttal process in our final paper.

---

### Official Review · Reviewer_Kgxg · 2023-07-06

**Soundness:** 3 good
**Presentation:** 3 good
**Contribution:** 3 good
**Rating:** 6
**Confidence:** 4

**Summary:**

This paper studies the problem of personalized federated learning with each client conducting dictionary learning on heterogeneous tasks. This paper splits the learned dictionary into a global dictionary and local dictionaries. It provides the conditions that two types of dictionaries can be provably identified. It designs a federated meta-algorithm where clients only pass estimated global dictionaries to the center. It also provides the linear convergence for the federated learning procedure under some assumptions.

**Strengths:**

1. The objective function is concise for the described personalized dictionary learning problem.
2. It gives conditions when global dictionaries and local dictionaries can be identified. The two conditions satisfy the intuition.
3. The algorithm finds the global dictionaries given initial client dictionaries by finding the shortest path based on DAG, which is an interesting solution.
4. The overall algorithm is theoretically guaranteed.
5. The writing of the paper is good. The description of the algorithm is clear.

**Weaknesses:**

The experiments are somewhat weak. Using each frame as a client is also strange in the third experiment.

**Questions:**

1. Please discuss the related techniques about graph methods when tackling the dictionary problem.
2. The dimension r_g of the global dictionary is the most important hyper-parameter of the algorithm. How to choose it in the experiments? How does the dimension r_g influence the algorithm? More discussions should be added.

**Limitations:**

More experiments should be added. Authors could take more experiments on tabular data sets. It is better to test how the dimension of the global dictionary r_g influences the performance.

---

> ### Author Rebuttal · Authors · 2023-08-09
>
> We are grateful for your insightful comments and suggestions.
>
> > The experiments are somewhat weak.
>
> We kindly refer the reviewer to our general response, where we have included numerical comparisons between our method and other existing methods in our global response. Our method indeed exhibits superior performances in all three case studies.
>
> > Using each frame as a client is also strange in the third experiment.
>
> We agree with the reviewer's observation that the utilization of each frame as a "client" is not a conventional approach. However, in this context, the definition of a "client" is somewhat artificial, driven by the ultimate objective of distinguishing common and individually distinctive elements, such as the background and the cars. In this framework, each "client" captures the unique features of a frame. It is worth noting that our definition of a client may also mirror real-world scenarios, where each client only has access to a limited number of frames that share local features (e.g., frames captured from a specific angle). Although such applications hold significance, they are not the primary focus of this paper.
>
> > Please discuss the related techniques about graph methods when tackling the dictionary problem.
>
> The authors thank the reviewer for this comment. On the theoretical studies of dictionary learning, graph-based methods have been used to provide early-stage estimations of the true dictionary, followed by classic alternating methods to achieve exact recovery. In *Arora et al.(2014)*, the authors provide a novel method for dictionary learning based on a connection graph, whose purpose is to detect whether or not two samples (or signals) share the same atom. Such an idea is further developed in *Agarwal et al.(2014)* and *Arora et al.(2015)*.
>
> The graph-based method in this paper, however, is fundamentally different from the previous works. The main idea behind our approach is to detect the commonalities and similarities among the atoms by casting it as a series of shortest path problems over a synthetically generated directed acyclic graph (DAG). To the best of our knowledge, such an approach has not been used before for identifying the common features in dictionary learning. The closest to our proposed method is the Federated
> Matched Averaging (FedMA) algorithm introduced by *Wang et al. (2020)* for the federated learning of neural network architectures. In this method, identifying a global model is cast as a series of assignment problems over bipartite graphs. This approach is a special case of our proposed graph-based method, where the number of layers is limited to two.
>
> Finally, to solve the shortest path problems, we used the well-known labeling algorithm, which can solve the single source shortest path problem in linear time (with respect to the number of edges in the graph). This algorithm is already implemented in the built-in MATLAB function "shortestpath.m".
>
> > The dimension $r_g$ of the global dictionary is the most important hyper-parameter of the algorithm. How to choose it in the experiments? How does the dimension $r_g$ influence the algorithm? More discussions should be added.
>
> We sincerely thank the reviewer for this very insightful comment. As the reviewer correctly mentioned, $r_g$ is an important hyper-parameter of our algorithm. A larger $r_g$ means more global atoms are sent between central servers and clients, which leads to a stronger collaboration between them. In the synthetic experiment, we assume to know the value of $r_g$, while in real-life applications, one needs to fine-tune this parameter. An interesting observation we made, which was omitted due to limited space, is that the proposed PerMA algorithm can be augmented by a simple detection mechanism for identifying the correct choice of $r_g$. Specifically, during the Global Matching step, where we iteratively remove shortest paths, we can closely monitor the length of the obtained shortest path. By terminating the removal of paths (i.e., adding global atoms) when the path's length experiences a significant increase beyond a predefined threshold, we can effectively identify the appropriate value of $r_g$ without requiring prior knowledge. This detection mechanism alleviates the burden of fine-tuning $r_g$ and allows for a more practical and robust implementation of the algorithm.
>
> To validate the efficacy of this approach, we conducted a series of experiments, the results of which are presented in Figure 3 in the pdf file. We use different $r_g = 4,6,8$ with $r=10$ and monitor the lengths of paths. As evident from the outcomes, a clear and drastic increase in the length of the $r_g+1$-th shortest path is observed, signifying the correct value of $r_g$.
>
> We will add the new experiments to the revised manuscript.
>
> ### Reference
>
> Agarwal, A., Anandkumar, A., Jain, P., Netrapalli, P., and Tandon, R. (2014). Learning sparsely used overcomplete dictionaries. In Conference on Learning Theory, pages 123–137. PMLR.
>
> Arora, S., Ge, R., Ma, T., and Moitra, A. (2015). Simple, efficient, and neural algorithms for sparse coding. In Conference on learning theory, pages 113–149. PMLR.
>
> Arora, S., Ge, R., and Moitra, A. (2014). New algorithms for learning incoherent and overcomplete dictionaries. In Conference on Learning Theory, pages 779–806. PMLR.
>
> Wang, H., Yurochkin, M., Sun, Y., Papailiopoulos, D., and Khazaeni, Y. (2020). Federated learning with matched averaging. arXiv preprint arXiv:2002.06440.209

---

> > ### Comment · Reviewer_Kgxg · 2023-08-16
> >
> > Thank the authors for their detailed rebuttal. The discussion on the related techniques and the way to choose the important hyperparameter address my issueses.

---

> > > ### Author Response · Authors · 2023-08-18
> > >
> > > We are pleased that our rebuttal is able to address your concern. We will integrate new experiments into our revised manuscript.

---

### Official Review · Reviewer_JdsU · 2023-07-25

**Soundness:** 3 good
**Presentation:** 3 good
**Contribution:** 3 good
**Rating:** 7
**Confidence:** 3

**Summary:**

This paper proposed a challenging problem named Personalized Dictionary Learning (PerDL), which learned a shared global dictionary and individual local dictionary for heterogeneous datasets.

In order to investigate the feasibility of the problem, several definitions and assumptions are provided to make the theoretical guarantee. Under these conditions, a meta-algorithm called Personalized Matchina and Averaging (PerMA) is proposed to solve the problem. The convergence of PerMA is theoretically guaranteed.

Experiments are conducted on synthetic, imbalanced digits reconstruction and video surveillance datasets, which show the effectiveness of PerMA.

**Strengths:**

(1). The problem is well-defined to ensure identifiability, feasibility, and convergence with the help of certain mild assumptions and definitions. This involves Assumptions 1 and 2, Definitions 1 and 2. This way, it is natural to investigate and derive a solution under the federated learning context.

(2). A federated meta-algorithm (PerMA) is proposed to solve the PerDL problem. In particular, Global Matching and Local Updates steps are designed in the federated setting. Global matching utilized a shortest path algorithm to tackle the non-convex and different initialization problems. Local updates employed a linearly-convergent algorithm.

(3). With proper assumption and mild conditions, the convergence of PerMA is proved, being a theoretical contribution to ensure the feasibility of the PerDL problem.

(4). Experiments on three settings verify the rationale of PerDL and the effectiveness of PerMA.

**Weaknesses:**

Overall, the theory and method is good.
The experiment is a bit weak, considering only an independent strategy is adopted as the baseline.
Can it be compared with other methods, such as personalized PCA?


**Questions:**

The comparison is a bit weak.
Is it possible to compare with other baselines in Dictionary learning or Federated learning?

---

> ### Author Rebuttal · Authors · 2023-08-09
>
> > Can it be compared with other methods, such as personalized PCA?/Is it possible to compare with other baselines in Dictionary learning or Federated learning?
>
> Thank you for this helpful suggestion. We kindly refer the reviewer to our general response, where we have included numerical comparisons between our method and other existing methods in our global response. Our method indeed exhibits superior performances in all three case studies.

---

> > ### Comment · Reviewer_JdsU · 2023-08-17
> >
> > Thanks for the rebuttal! The included numerical comparisons addressed my concern.
> > I will keep my score as Accept.

---

> > > ### Author Response · Authors · 2023-08-18
> > >
> > > We are pleased that our rebuttal is able to address your concern and thank you for your support for the paper!

---

### Author Rebuttal · Authors · 2023-08-09

We are thankful to the reviewers for carefully reading and commenting on the strengths and weaknesses of our paper. A recurring comment among the reviewers was on the limitation of our experiments. We have thoroughly addressed this comment by conducting more experiments on our method and comparing its performance with three methods (suggested by the reviewers). The results can be found in the uploaded pdf file. We upload the code anonymously via [this link](https://anonymous.4open.science/r/PerMA-4B9E/README.md). We will integrate the results into the final version of our paper. In what follows, we briefly explain our new experiments.

### Synthetic Dataset

In this section, we compare our method with *Gkillas et al. (2022)* and *Huang et al. (2022)* under the same setting as in Section 5.1. The results can be seen in Figure 1. In the first row, clients are provided with heterogeneous datasets with similar sizes; in the second row, we consider the special cases in which one of the clients has an insufficient sample size and evaluate the performance of the dictionary learned by that specific client. The first column corresponds to final errors with varying $d$; the second column corresponds to total running times with varying $d$; the third column corresponds to final errors with varying $N$; and the fourth column corresponds to final errors with varying sparsity levels. All the results are averaged over 3 independent trials. As it can be seen in Figure 1, PerMA consistently outperforms methods proposed by *Gkillas et al. (2022)* and *Huang et al. (2022)*.

### MNIST Dataset

Thanks to the suggestions by Reviewer zsPp, we compare the quality of the reconstructed images using dictionaries learned from different methods under three metrics: MSE, PSNR and SSIM. A smaller MSE, a larger PSNR, and a larger SSIM indicate better image reconstruction quality. In Table 1, $k$ denotes the number of atoms used to reconstruct the image.
As can be seen in the table, PerMA achieves the best result in all sections except for the training time.

### Surveillance Video Dataset

In this paper, we use PerMA on the surveillance video datasets, with the goal of separating common elements shared by all clients (the background) and unique elements (different cars). Such a task cannot be accomplished by *Gkillas et al. (2022)* and *Huang et al. (2022)* due to their lack of personalization. We refer interested reviewers to our response to Reviewer zsPp for further discussion of PerMA on video surveillance.  As a result, to compare our method with the state-of-the-art, we run Personalized PCA (PerPCA) introduced by *Shi and Kontar (2022)* on the same datasets. According to Figure 2, PerDL outperforms PerPCA by achieving better separation and higher resolution. We notice that Reviewer zsPp's suggestion to use F-score to quantify our result. However, F-score is mainly used as a metric to evaluate the accuracy of object detection strategy. Turning PerMA into an object detection strategy is an interesting future research direction but is not the focus of our paper.

### Reference
Gkillas, A., Ampeliotis, D., and Berberidis, K. (2022). Federated dictionary learning from non-iid data. In 2022 IEEE 14th Image, Video, and Multidimensional Signal Processing Workshop(IVMSP), pages 1–5. IEEE.

Huang, K., Liu, X., Li, F., Yang, C., Kaynak, O., and Huang, T. (2022). A federated dictionary learning method for process monitoring with industrial applications. IEEE Transactions on Artificial Intelligence.

Shi, N. and Kontar, R. A. (2022). Personalized pca: Decoupling shared and unique features. arXiv preprint arXiv:2207.08041.

---

### Comment · Area_Chair_LvpJ · 2023-08-13
**Post-rebuttal review**

Dear Reviewers,

Thanks for your time and efforts in reviewing this paper. The author rebuttal is now available. Please read the rebuttal and update your comments ASAP. Thank you very much!

Best,
AC

---

### Decision · Program_Chairs · 2023-09-21

**Decision:**

Accept (poster)

**Comment:**

The paper receives four informative reviews. After the author rebuttal, the majority of the reviews agree that the paper is contributive and suggest acceptance. Reviewer ED5i, who didn’t response to the authors’ rebuttal, remains negative. As far as I can see, the authors’ rebuttal has addressed properly Reviewer ED5i's concerns about the experimental comparisons. Thus, I would recommend accepting the paper.